behaviour

long-wavelength sensitive, cone opsin, spectral sensitivity, optomotor response, near-infrared light, fish

**Authors for correspondence:**
Megumi Matsuo
e-mail: mmatuso@fc.jwu.ac.jp
Shoji Fukamachi
e-mail: fukamachi@fc.jwu.ac.jp

# Behavioural red-light sensitivity in fish according to the optomotor response

Megumi Matsuo[1], Yasuhiro Kamei[2,3] and Shoji Fukamachi[1]

[1] Laboratory of Evolutionary Genetics, Department of Chemical and Biological Sciences, Japan Women's University, Tokyo 112-8681, Japan
[2] Spectrography and Bioimaging Facility, National Institute for Basic Biology, Aichi 444-8585, Japan
[3] Department of Basic Biology, School of Life Science, The Graduate University for Advanced Studies (SOKENDAI), Aichi 444-8585, Japan

MM, 0000-0003-2053-0271; YK, 0000-0001-6382-1365; SF, 0000-0001-7967-1883

Various procedures have been adopted to investigate spectral sensitivity of animals, e.g. absorption spectra of visual pigments, electroretinography, optokinetic response, optomotor response (OMR) and phototaxis. The use of these techniques has led to various conclusions about animal vision. However, visual sensitivity should be evaluated consistently for a reliable comparison. In this study, we retrieved behavioural data of several fish species using a single OMR procedure and compared their sensitivities to near-infrared light. Besides cavefish that lack eyes, some species were not appropriate for the OMR test because they either stayed still or changed swimming direction frequently. Eight of 13 fish species tested were OMR positive. Detailed analyses using medaka, goldfish, zebrafish, guppy, stickleback and cichlid revealed that all the fish were sensitive to light at a wavelength greater than or equal to 750 nm, where the threshold wavelengths varied from 750 to 880 nm. Fish opsin repertoire affected the perception of red light. By contrast, the copy number of long-wavelength-sensitive (*LWS*) genes did not necessarily improve red-light sensitivity. While the duplication of *LWS* and other *cone opsin* genes that has occurred extensively during fish evolution might not aid increasing spectral sensitivity, it may provide some other advantageous ophthalmic function, such as enhanced spectral discrimination.

## 1. Introduction

In the outer retinae of vertebrates, there are two classes of light-sensitive cells, rod photoreceptors and cone photoreceptors. Rod

**Figure 1.** Fish species in this work. (*a*) Phylogenetic relationships of fish. A tree was created based on previous works [26–28]. (*b*) Visual sensitivity of fish. The absorption spectra of visual pigments (upper row) and electrophysiological and behavioural response (lower row) of fish are summarized. A horizontal axis shows the wavelength of light. The absorption spectrum of SWS1 is depicted in violet, SWS2 in blue, RH2 in green, LWS in red and RH1 in black. A bar indicates the longest wavelength that has provoked a fish response; phototaxis (green), ERG (orange), OMR (blue) and electrophysiological response at the optic nerve and cardiac conditioned response (black). The longest wavelengths to which fish responded in this work are shown as magenta bars. Guppy has one additional LWS pigment with unknown absorption spectrum. Guppy LWS-1 is LWS-1/180Ser. Mbuna has one LWS pigment with unknown absorption spectrum. All the chromophores are A1, but those of goldfish are A2. The original references on which this figure is based are given in electronic supplementary material, table S1.

photoreceptor cells are responsible for scotopic vision, the vision working under dim light. Photopic vision, the vision working under daylight, is mediated by cone photoreceptors. Colour information is mainly gained by comparing the output of cones [1]. Rods and cones contain visual pigments that are composed of a light-absorbing chromophore linked to a protein, opsin. Rods of most vertebrates contain a single rod opsin (rhodopsin; RH1). Some deep-sea teleost lineages, such as Myctophidae, Stylephoridae and Diretmidae, express multiple rod opsins [2]. Amphibia have two types of rods, typical rods and green rods whose absorbance peaks in the blue part of the spectrum [3]. Individual cones usually contain a single type of visual pigment with different absorption maxima ($\lambda_{max}$). There are four types of cone opsin: ultraviolet sensitive (SWS1), short-wavelength sensitive (SWS2), medium-wavelength sensitive (RH2) and medium-to-long-wavelength sensitive (MWS/LWS) opsin. In trout, cichlid and anemonefish, some photoreceptor cells co-express spectrally distinct opsins [4–8]. Most fish species have more than one spectral cone type [9–17].

Using a wooden model almost 70 years ago, Tinbergen [18] showed that the colour red triggered aggression in male stickleback fish. Red is also a nuptial colour [19,20] and influences mating preference in fish [21,22]. Previously using medaka with the *LWS* gene knocked out, we showed that changes in red-light sensitivity affect mate choice [23–25]. Red-light sensitivity can also be important for other fish species. To understand the visual sensitivity of several fish species to red light, we conducted an optomotor response (OMR) assay in this study.

The OMR is a classic behavioural response that exploits the tendency of an animal to follow a moving pattern and is employed to study visual sensitivities. Besides the OMR, fish vision has been examined in several ways, including electroretinography (ERG), phototaxis and optokinetic response (OKR). As far as we know, we summarized the longest wavelength to which animals responded (figure 1; electronic supplementary material, table S1). In previous studies, guppy responded to light of 600 nm as

measured using the OMR assay [29]; goldfish, 699 nm [30]; zebrafish, 699 nm [31]; medaka, 830 nm [24,25] and Nile tilapia, 780 nm [32]. Other techniques have also revealed the visual capabilities of fish. Based on ERG data, zebrafish responded to 640 nm [33] and three-spined stickleback to 670 nm light [34]. In Nile tilapia, the cardiac conditioning method has shown these fish respond to light of 865 nm [35]. Using rainbow trout, optic nerve recordings revealed that light of 640 nm induced fish response [36]. Phototaxis has been observed in near-infrared (NIR) light in zebrafish, guppy and Nile tilapia. Phototaxis was induced by light of 910 nm in zebrafish and guppy and of 930 nm in Nile tilapia [37]. As mentioned above, different techniques have yielded different results and led to a variety of conclusions. To assess the impact of the difference in opsin repertoire on visual sensitivity of deep-red and NIR light, we conducted OMR assays based on a single procedure.

To study spectral sensitivity to red light, we used medaka, goldfish, zebrafish, guppy, three-spined stickleback, cichlid (Nile tilapia and mbuna), bronze corydoras, Mexican cavefish, glowlight tetra, Japanese striped loach, rainbow trout and Senegal bichir (figure 1). All the fish species studied in this work belong to the class Actinopterygii. Mexican cavefish, also known as blind cave tetra, have degenerated eyes and have lost schooling behaviour [38]. We introduced Mexican cavefish as a negative control in this study.

## 2. Methods

### 2.1. Animals

The experiments were conducted following the Guidelines for Proper Conduct of Animal Experiments and approved by the National Institute for Basic Biology (NIBB), Aichi, Japan (18A093).

Medaka (*Oryzias latipes*) were reared in our laboratory under a 14/10 h light/dark cycle. Fish were maintained under ordinary fluorescent light (figure 2). All the other fish species were hatchery reared and purchased from the local fish suppliers. These included goldfish (*Carassius auratus*), zebrafish (*Danio rerio*), guppy (*Poecilia reticulata*), three-spined stickleback (*Gasterosteus aculeatus*), mbuna (*Metriaclima zebra*), glowlight tetra (*Hemigrammus erythrozonus*), Mexican cavefish (*Astyanax mexicanus*), Nile tilapia (*Oreochromis niloticus*), rainbow trout (*Oncorhynchus mykiss*), bronze corydoras (*Corydoras aeneus*), Japanese striped loach (*Cobitis biwae*) and Senegal bichir (*Polypterus senegalus*). The fish species used in this work are summarized in figure 1. Fish tanks were maintained under the identical light/ dark cycle as medaka until the experiment. We measured the total length (TL) of the fish from the snout to the caudal fin. All the fish were 3–5 cm long except for Senegal bichir and rainbow trout. Senegal bichir was 7 cm long and the TL of rainbow trout was 10 cm. Six each of medaka, goldfish, zebrafish, guppy, three-spined stickleback, mbuna, Mexican cavefish and Nile tilapia were subjected to the OMR test. One each of Senegal bichir, Japanese striped loach and rainbow trout, two glowlight tetra and three bronze corydoras were used.

### 2.2. Behavioural test

To assess red-light sensitivity, we performed an OMR assay in deep-red and NIR light based on the procedure previously described [25]. Individual fish were placed in a cylindrical glass tank with a diameter of 18.5 cm. Around the aquarium, a drum with a diameter of 24 cm rotated black and white stripes (2 cm wide). The speed of the rotation was 10 r.p.m. ($60° \, s^{-1}$). The tank was irradiated from above using monochromatic light generated by an Okazaki large spectrograph at the National Institute for Basic Biology (Aichi, Japan) [39]. In the centre of the tank, a 50 ml centrifuge tube filled with water was placed to prevent fish from short-cutting. As fish swam freely in the aquarium, the distance between fish and the striped pattern changed during behavioural assays, with a minimum of 5.5 cm. Fish were light adapted for more than 2 h before the experiments. The wavelength was shifted from a shorter to longer wavelength. The spectra of all wavelengths used in behavioural assays and ambient light are shown in figure 2. The intensity and spectra of light were measured using a spectroradiometer (S-2440C; Soma Optics, Tokyo, Japan, table 1). The photon density used in this study (table 1) was different from our previous paper [25] because the photon density changed depending on the cumulative lighting time of the light source (the xenon lamp).

Before all experiments, we put fish in separate cups and kept them isolated until the end of all the behavioural tests. First, a fish was gently transferred with water from the cup to the testing apparatus. We checked the OMR apparatus, opened the shutter of the monochromatic light source, started video recording and then turned all ceiling lights off. We counted 30 s for acclimatization without rotating

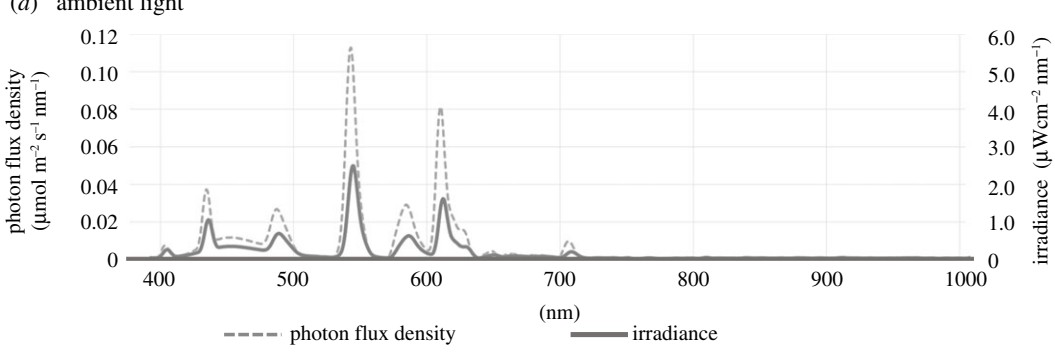

(a) ambient light

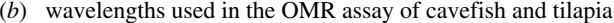

(b) wavelengths used in the OMR assay of cavefish and tilapia

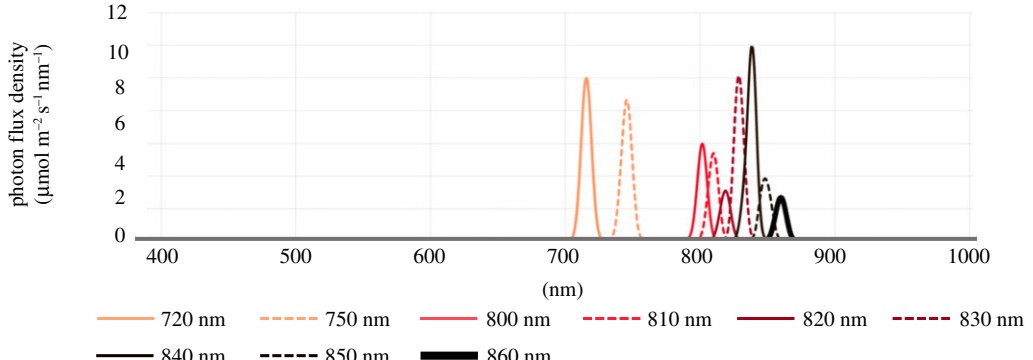

(c) wavelengths used in the first-round OMR assay

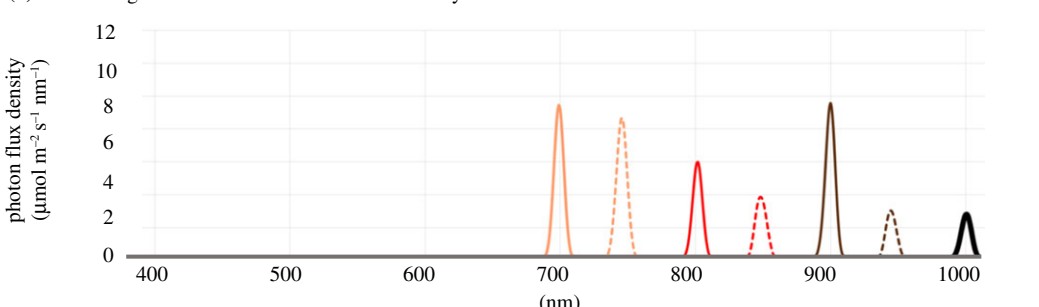

(d) wavelengths used in the second-round OMR assay

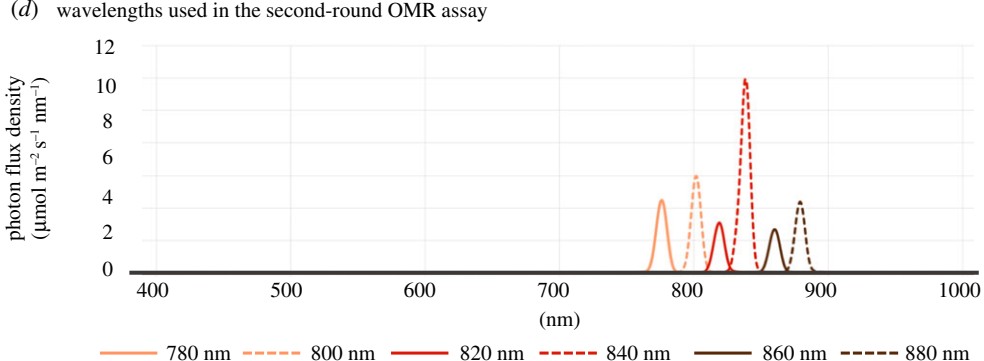

**Figure 2.** Ambient light and monochromatic light in this work. Intensity and spectra of light were measured using a spectroradiometer. (a) Intensity and spectrum of ambient light. (b–d) Spectra of monochromatic light used in the OMR assay. We performed behavioural assays under the light of 700, 720, 750, 780, 800, 810, 820, 830, 840, 850, 860, 880, 900, 950 and 1000 nm. At all points, the spectra were measured every 1 nm and are shown differently coloured. Spectra of light used in OMR assays of cavefish and tilapia (b), the first round (c) and the second round (d) are illustrated.

**Table 1.** Monochromatic light in the aquarium of OMR testing apparatus.

| peak wavelength (nm) | 700 | 750 | 780 | 800 | 810 | 820 | 830 | 840 | 850 | 860 | 880 | 900 | 950 | 1000 |
|---|---|---|---|---|---|---|---|---|---|---|---|---|---|---|
| photon flux density ($\mu$mol m$^{-2}$ s$^{-1}$) | 17 | 14 | 10 | 11 | 10 | 9.5 | 12 | 15 | 9 | 8.6 | 19 | 23 | 43 | 17 |

the stripes of the OMR apparatus, followed by the subsequent OMR test. During the behavioural test, the drum rotated, switching direction every 30 s three times. After stopping the rotation, we turned on the ceiling light, closed the shutter of the monochromatic light source and stopped video recording. After the test, fish were light adapted again. We used an infrared camera to record the behaviour of the fish (A10FHDIR; Kenko, Tokyo, Japan).

We conducted two rounds of OMR assay. The first round was conducted to narrow the range of wavelength of detailed investigation to be conducted in the second-round test. In the first-round behavioural assays, fish were tested under the light at 50 nm wavelength intervals from 700 to 1000 nm. In the second round, we checked fish behaviour under the light at 20 nm intervals from 780 to 880 nm. The approximate schedule of experiments is described in electronic supplementary material, table S2. In guppy, all four parameters (see below) fell at 850 nm. Therefore, we skipped 900 and 1000 nm in the first-round OMR assay, and focused around 840 and 860 nm in the second-round OMR test (780, 820, 840, 860 and 880 nm), due to the schedule of the experiment. In medaka, previous studies showed that these fish are not OMR positive under light of wavelength greater than 830 nm [24,25]. We did not test 900 or 1000 nm in the first round and focused on 820 and 840 nm in the second round (780, 800, 820, 840 and 880 nm). All behaviour during the OMR test was recorded as a video file. In some cases, we modified the video file by filtering the colour or adjusting the luminance to exclude noise. For example, tracking was made difficult when a male guppy's long tail fin disturbed the water surface. By applying a grey filter and modifying the overall luminance of the recorded video, the motion of the guppy male was tracked successfully.

The data were converted into $x$–$y$ coordinates by UMATracker software [25,40]. With sets of coordinates, we calculated four parameters: (i) Delay, the elapsed time (s) until a fish started to follow the pattern after switching the rotating direction of the drum; (ii) Duration, the ratio of the time during which a fish followed the striped pattern divided by the total testing time; (iii) Angular velocity, the speed in degrees per second at which a fish swam to chase the stripes; and (iv) Distance, the overall swimming distance (rounds) a fish swam in the direction of the rotating apparatus. When a fish swam in the opposite direction, that distance was subtracted from the overall distance for the trial.

## 2.3. Statistics

Datasets of the four parameters (see above) were analysed with R statistical software for Windows v. 3.2.0 [41,42] and 'anovakun' v. 4.6.2, an analysis of variance (ANOVA) function that runs on R software (http://riseki.php.xdomain.jp/index.php?FrontPage), based on the work of Donoghue [43] and Rasmussen [44]. First, a Bartlett test was conducted to assess the homogeneity of variances. For the datasets with equal variances, parametric procedures were applied for further analyses. Alternatively, data were subjected to non-parametric statistical procedures. Parametric statistical analyses included Mendoza's multisample sphericity test, one-way repeated-measures ANOVA and Shaffer's modified sequentially rejective Bonferroni procedure. When the sphericity test yielded a significant $p$-value, further analyses were adjusted using the epsilon (Greenhouse–Geisser). For non-parametric data, Steel–Dwass test or Steel test was conducted (http://CRAN.R-project.org/package=kSamples). In this paper, the $p$-value was from the data either of Shaffer's modified sequentially rejective Bonferroni procedure or Steel–Dwass test unless otherwise noted.

# 3. Results

## 3.1. Five fish species did not have an optomotor response

First, we performed an OMR assay at 700 or 720 nm. The response of four fish species (rainbow trout, Nile tilapia, Senegal bichir and Japanese striped loach) suggests they should be evaluated for their photosensitivity by other methods. For example, young rainbow trout and Nile tilapia turned around frequently during the OMR test, regardless of the rotating stripes (electronic supplementary material,

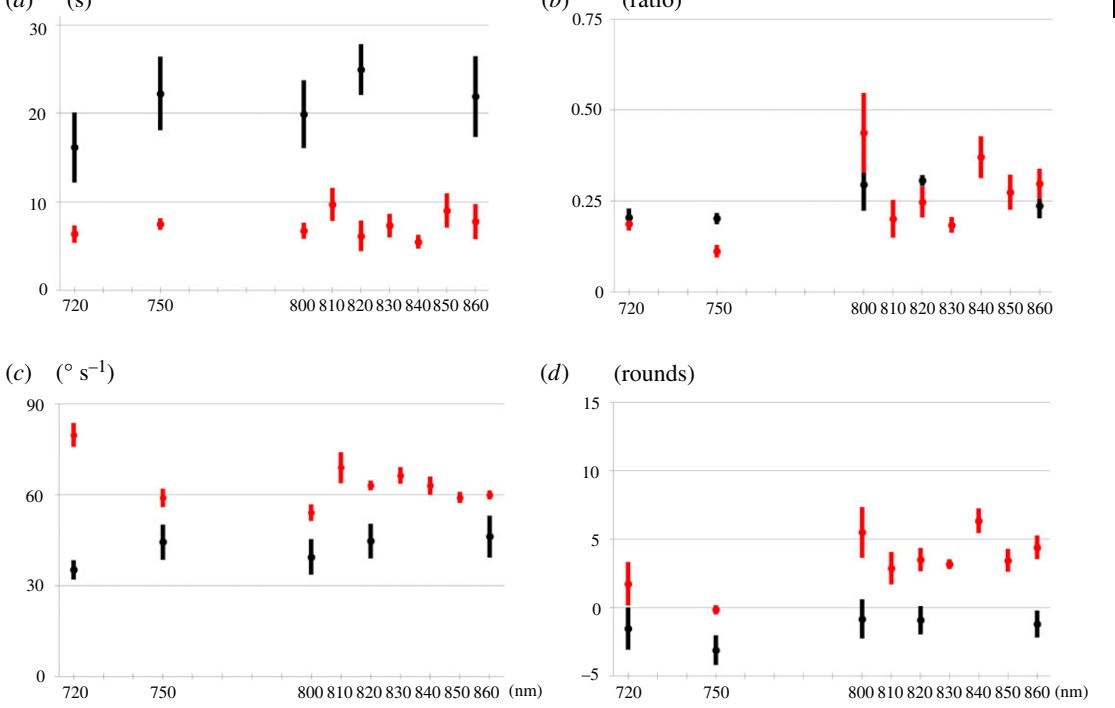

**Figure 3.** Behavioural response of Nile tilapia and Mexican cavefish under the light from 720 to 860 nm. Behavioural tests were conducted using Nile tilapia and Mexican cavefish. In all figures, the horizontal axis indicates the wavelength. The OMR was quantified using the four parameters (refer to '§2.2' for details of the four parameters). (*a*) Delay, (*b*) Duration, (*c*) Angular velocity and (*d*) Distance. Values represent means. Error bars are s.e.

video file S1). Japanese striped loach and Senegal bichir seldom swam and did not follow the stimuli. Yet, the bichir moved its head whenever we switched the direction of stripe rotation (electronic supplementary material, video file S2), suggesting that it perceived light even though the OMR was negative.

Mexican cavefish have degenerated eyes. We tested their behaviour under light of wavelengths 720, 750, 800, 820 and 860 nm. Values of the four parameters were computed and are summarized in figure 3. Under all wavelengths, the Delay parameter fluctuated and had no significant difference (one-way repeated-measures ANOVA, $F_{4,20} = 0.6431$, $p = 0.6381$). Duration was between 0.202 (750 nm) and 0.304 (820 nm) ($p > 0.05$ in all comparisons). From 720 to 860 nm, Angular velocity was maintained (one-way repeated-measures ANOVA, $F_{4,20} = 0.7193$, $p = 0.5888$). Distance at any wavelength was negative, which meant cavefish never followed the signals (one-way repeated-measures ANOVA, $F_{4,20} = 0.5802$, $p = 0.6804$). Based on these four parameters, we judged cavefish to be OMR negative at all wavelengths tested.

Nile tilapia did not exhibit an obvious OMR. However, Delay increased towards 860 nm, Angular velocity was about $60°\,s^{-1}$, and Duration and Distance fluctuated but never declined (figure 3). When comparing the behavioural response of cavefish and tilapia, Nile tilapia could perceive light of wavelengths between 720 and 860 nm.

## 3.2. Eight fish species were optomotor response positive

Based on our manual observations, eight fish species (medaka, goldfish, zebrafish, guppy, three-spined stickleback, mbuna, glowlight tetra and bronze corydoras) were OMR positive (examples of the OMR assay of tetra and corydoras are in electronic supplementary material, video files S3 and S4). Among them, six fish species (medaka, goldfish, zebrafish, guppy, three-spined stickleback and mbuna), widely used as model animals, were subjected to further behavioural assays.

## 3.3. Deep-red and NIR light elicited an optomotor response in fish

Medaka, goldfish, zebrafish, guppy, three-spined stickleback and mbuna followed the rotating striped pattern at 700 nm. We conducted OMR assays using these six fish species under light of wavelengths between 700 and 1000 nm (first-round OMR test, figure 4).

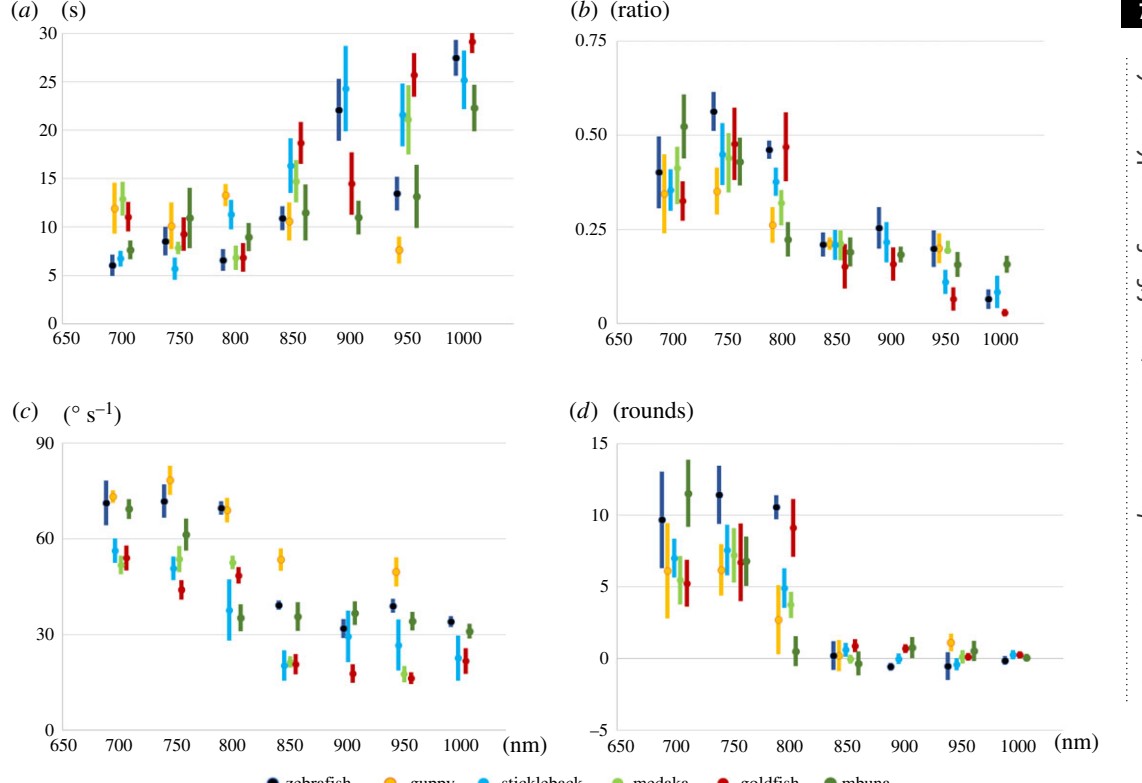

**Figure 4.** Behavioural response of medaka, goldfish, zebrafish, guppy, three-spined stickleback and mbuna in the first-round OMR assay. Behavioural tests were conducted using six fish species. In all figures, the horizontal axis indicates the wavelength. The OMR was quantified using the four parameters. (*a*) Delay, (*b*) Duration, (*c*) Angular velocity and (*d*) Distance. Values represent means. Error bars are s.e. We summarized the graphs according to fish species in electronic supplementary material, figure S1.

We calculated four parameters to quantify each fish's response: Delay, Duration, Angular velocity and Distance (figure 4 and table 2). Standard errors of the Delay parameter were rather large, fluctuated with wavelength and seemed to depend on chance. A sharp drop in Duration, Angular velocity and Distance indicated a loss of OMR behaviour. When an animal gradually lost its vision, Delay increased while Duration, Angular velocity and Distance decreased.

The Duration of six fish species at 700 nm ranged from 0.325 to 0.523. This value was attenuated towards longer wavelengths. At 700 nm, zebrafish followed stripes at the ratio of 0.40 and 0.06 at 1000 nm (800 > 1000: $t = 21.3510$, $p = 0.0047$). The Duration of goldfish was the same as that of zebrafish. At 700 nm, the ratio was 0.35 and 0.02 at 1000 nm ($p > 0.05$). In medaka, guppy and stickleback, the highest ratio was 0.44, 0.35 and 0.45 at 750 nm, respectively. The Duration for these three fish species was attenuated with longer wavelength (stickleback, one-way repeated-measures ANOVA, $F_{2.45,12.26} = 8.1019$, $p = 0.0042$; medaka, 800 > 950: $t = 2.8823$, $p = 0.0322$; guppy, $p > 0.05$ in all comparisons). In mbuna, Duration at 700 and 750 nm was 0.52 and 0.43, respectively, dropped at 800 nm and showed little change up to 1000 nm.

At 700 nm, fish adjusted their speed to rotating stripes, around $60° s^{-1}$. And the Angular velocity of mbuna fell at 800 nm (750 > 800: $t = 19.0225$, $p = 0.0001$). That of the other five fish decreased towards 800 nm. The $p$-value of one-way repeated-measures ANOVA of guppy was 0.0097 ($F_{4,12} = 11.0937$). Medaka had a significant $p$-value for Angular velocity (800 > 850: $t = 21.5642$, $p = 0.0000$); goldfish (750 > 900: $t = 29.9377$, $p = 0.0017$); stickleback (750 > 850: $t = 7.4728$, $p = 0.0102$).

Zebrafish swam 11.42 rounds at 750 nm and 10.56 rounds at 800 nm. The Distance of zebrafish dropped at 850 nm ($p > 0.05$). Goldfish followed the stripes and swam 9.11 rounds at 800 nm. The Distance of goldfish dropped at 850 nm and kept decreasing as the wavelength approached 950 nm ($p > 0.05$). Medaka, guppy and stickleback swam most at 750 nm (7.20, 6.18 and 7.57 rounds, respectively). Their Distance dropped at 850 nm (medaka, 800 > 850: $t = 2.8823$, $p = 0.0322$; sticklebacks and guppy, $p > 0.05$ in all comparisons).

**Table 2.** Statistics of four parameters of medaka, goldfish, zebrafish, guppy, three-spined stickleback and mbuna in the first round. The four parameters (Delay, Duration, Angular velocity, Distance) of OMR were statistically analysed by either parametric or non-parametric methods. The wavelength combinations that showed significant differences are tabulated along with the $p$-values. n.s.: $p > 0.05$, not significant.

### medaka

| | | Delay | Duration | | Angular velocity | | Distance | |
|---|---|---|---|---|---|---|---|---|
| | | p-value | p-value | wavelength | p-value | wavelength | p-value | wavelength |
| parametric | one-way repeated-measures ANOVA ($p$) | 0.0008 | | | 0.0000 | | | |
| | Shaffer's modified | n.s. | | | 0.0000 | 800 > 850 | | |
| | sequentially rejective | | | | 0.0001 | 700 > 850 | | |
| | Bonferroni procedure | | | | 0.0003 | 750 > 950 | | |
| | (adjusted $p$) | | | | 0.0010 | 700 > 950 | | |
| | | | | | 0.0022 | 800 > 950 | | |
| | | | | | 0.0033 | 750 > 850 | | |
| non-parametric | Steel–Dwass test | | 0.03225 | 700 : 950 | | | 0.03225 | 700 : 850 |
| | | | 0.03225 | 750 : 950 | | | 0.03225 | 750 : 850 |
| | | | 0.03225 | 800 : 950 | | | 0.03225 | 750 : 950 |
| | | | | | | | 0.03225 | 800 : 850 |

### goldfish

| | | Delay | | Duration | Angular velocity | | Distance |
|---|---|---|---|---|---|---|---|
| | | p-value | wavelength | p-value | p-value | wavelength | p-value |
| parametric | one-way repeated-measures ANOVA ($p$) | 0.0000 | | | 0.0000 | | |
| | Shaffer's modified | 0.0006 | 750 < 950 | | 0.0017 | 750 > 900 | |
| | sequentially rejective | 0.0114 | 750 < 1000 | | 0.0138 | 750 > 950 | |
| | Bonferroni procedure | 0.0294 | 700 < 1000 | | 0.0496 | 700 > 950 | |
| | (adjusted $p$) | | | | | | |
| non-parametric | Steel–Dwass test | | | n.s. | | | n.s. |

### zebrafish

| | | Delay | | Duration | | wavelength | Angular velocity | Distance |
|---|---|---|---|---|---|---|---|---|
| | | p-value | wavelength | p-value | wavelength | | p-value | p-value |
| parametric | one-way repeated-measures ANOVA ($p$) | 0.0000 | | 0.0037 | | | 0.0000 | |
| | Shaffer's modified | 0.0155 | 700 < 950 | 0.0047 | 800 > 1000 | | | |
| | sequentially rejective | 0.0155 | 700 < 1000 | | | | | |
| | Bonferroni procedure | 0.0197 | 750 < 1000 | | | | | |
| | (adjusted $p$) | | | | | | | |
| non-parametric | Steel–Dwass test | | | | | | n.s. | n.s. |

### guppy

| | | Delay | Duration | Angular velocity | |
|---|---|---|---|---|---|
| | | p-value | p-value | p-value | |
| parametric | one-way repeated-measures ANOVA ($p$) | 0.2721 | | 0.0097 | |
| | Shaffer's modified | — | | n.s. | |
| | sequentially rejective | | | | |
| | Bonferroni procedure | | | | |
| | (adjusted $p$) | | | | |
| non-parametric | Steel–Dwass test | | n.s. | | |

### stickleback

| | | Distance | Duration | | Delay | Angular velocity | | wavelength |
|---|---|---|---|---|---|---|---|---|
| | | p-value | p-value | | p-value | p-value | wavelength | |
| parametric | one-way repeated-measures ANOVA ($p$) | | 0.0042 | | | 0.0012 | | |
| | Shaffer's modified | | n.s. | | | 0.0100 | 700 > 850 | |
| | sequentially rejective | | | | | 0.0102 | 750 > 850 | |
| | Bonferroni procedure | | | | | 0.0447 | 700 > 1000 | |
| | (adjusted $p$) | | | | | | | |
| non-parametric | Steel–Dwass test | n.s. | | | n.s. | | | |

### mbuna

| | | Delay | | Duration | Angular velocity | | wavelength | Distance |
|---|---|---|---|---|---|---|---|---|
| | | p-value | wavelength | p-value | p-value | wavelength | | p-value |
| parametric | one-way repeated-measures ANOVA ($p$) | 0.0430 | | | 0.0000 | | | |
| | Shaffer's modified | 0.0285 | 800 < 1000 | | 0.0001 | 700 > 950 | 700 > 900 | |
| | sequentially rejective | 0.0452 | 700 < 1000 | | 0.0001 | 700 > 1000 | 750 > 950 | |
| | Bonferroni procedure | | | | 0.0001 | 750 > 800 | 750 > 1000 | |
| | (adjusted $p$) | | | | 0.0002 | 750 > 850 | 750 > 900 | |
| | | | | | 0.0004 | 700 > 850 | | |
| | | | | | 0.0006 | 700 > 800 | | |
| non-parametric | Steel–Dwass test | | | n.s. | | | | n.s. |

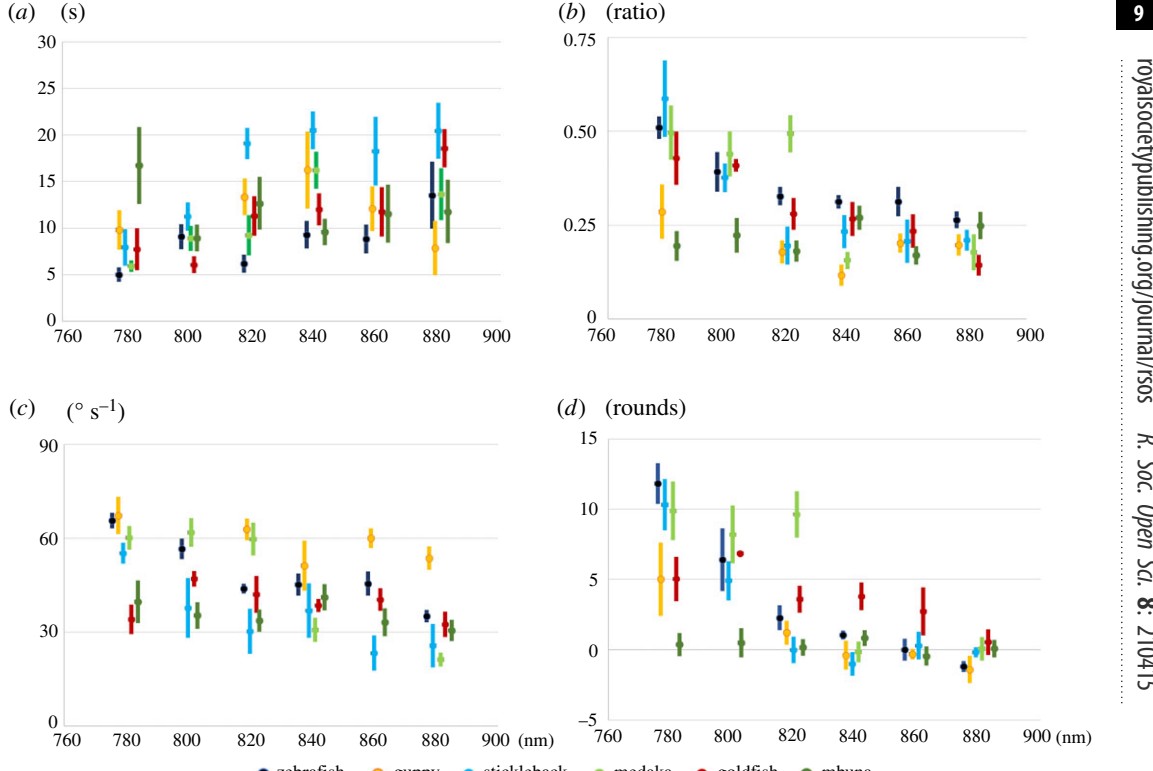

**Figure 5.** Behavioural response of medaka, goldfish, zebrafish, guppy, three-spined stickleback and mbuna in the second-round OMR assay. Behavioural tests were conducted using six fish species. In all figures, the horizontal axis indicates the wavelength. The OMR was quantified by the four parameters. (*a*) Delay, (*b*) Duration, (*c*) Angular velocity and (*d*) Distance. Values represent means. Error bars are s.e. We summarized the graphs according to fish species in electronic supplementary material, figure S2.

Among the four parameters, the Distance of six fish species converged around zero and was informative when judging their behaviour. When fish gradually lost their vision, all fish species swam about zero in total in behavioural tests. Results of the four parameters suggest that behavioural sensitivity of six fish species changed roughly around 800 or 850 nm. Accordingly, we next focused on the wavelengths from 780 to 880 nm.

## 3.4. Threshold wavelength of visible red light was estimated

The OMR test was conducted on six fish species under light of wavelengths ranging from 780 to 880 nm at 20 nm intervals (second-round OMR test, figure 5 and table 3). All the individuals tested were different fish from those of first-round OMR. Derived behavioural data were analysed using a repeated-measures design (for details, refer to Methods).

### 3.4.1. Medaka

Delay yielded a *p*-value of 0.0108 (one-way repeated-measures ANOVA, $F_{4,16} = 4.6771$). Values at 820 nm of Duration, Angular velocity and Distance differed from those at 840 nm ($t = 8.2527$, $p = 0.0118$; $t = 5.8632$, $p = 0.0169$; $t = 6.2946$, $p = 0.0325$, respectively). Delay increased and Duration, Angular velocity and Distance decreased toward 840 nm. Behavioural sensitivity changed between 820 and 840 nm, indicating that the threshold wavelength of visible light was between the range of 820 and 840 nm.

### 3.4.2. Zebrafish

Delay did not indicate any statistical trend in all the wavelengths tested. Based on statistics of Duration and Angular velocity, values at 780 nm were significantly different from those at longer wavelengths ($780 > 840$, $t = 6.0189$, $p = 0.0182$ in Duration; $780 > 820$, $t = 9.6012$, $p = 0.0031$ in Angular velocity). The

**Table 3.** Statistics of four parameters of medaka, goldfish, zebrafish, guppy, three-spined stickleback and mbuna in the second round. The four parameters (Delay, Duration, Angular velocity, Distance) of OMR were statistically analysed by either parametric or non-parametric methods. The wavelength combinations that showed significant differences are tabulated along with the $p$-values. n.s.: $p > 0.05$, not significant.

**medaka**

|  | Delay p-value | Duration p-value | Duration wavelength | Angular velocity p-value | Angular velocity wavelength | Distance p-value | Distance wavelength |
|---|---|---|---|---|---|---|---|
| **parametric** |  |  |  |  |  |  |  |
| one-way repeated-measures ANOVA ($p$) | 0.0108 | 0.0001 |  | 0.0000 |  | 0.0001 |  |
| Shaffer's modified sequentially rejective | n.s. | 0.0118 | 820 > 840 | 0.0018 | 780 > 880 | 0.0325 | 820 > 840 |
| Bonferroni procedure (adjusted $p$) |  | 0.0269 | 780 > 840 | 0.0048 | 800 > 880 | 0.0365 | 820 > 880 |
|  |  | 0.0311 | 800 > 840 | 0.0067 | 820 > 880 | 0.0423 | 780 > 840 |
|  |  |  |  | 0.0093 | 780 > 840 |  |  |
|  |  |  |  | 0.0145 | 800 > 840 |  |  |
|  |  |  |  | 0.0169 | 820 > 840 |  |  |

**goldfish**

|  | Delay p-value | Delay wavelength | Duration p-value | Duration wavelength | Angular velocity p-value | Angular velocity wavelength | Distance p-value | Distance wavelength |
|---|---|---|---|---|---|---|---|---|
| **parametric** |  |  |  |  |  |  |  |  |
| one-way repeated-measures ANOVA ($p$) | 0.0034 |  | 0.0008 |  | 0.0256 |  | 0.0000 |  |
| Shaffer's modified sequentially rejective | 0.0068 | 800 < 880 | 0.0164 | 800 > 880 | 0.0316 | 800 > 880 | 0.0139 | 780 > 880 |
| Bonferroni procedure (adjusted $p$) |  |  |  |  |  |  | 0.0139 | 780 > 820 |
|  |  |  |  |  |  |  | 0.0159 | 800 > 840 |
|  |  |  |  |  |  |  | 0.0357 | 800 > 860 |
|  |  |  |  |  |  |  | 0.0415 | 780 > 860 |
|  |  |  |  |  |  |  | 0.0467 | 780 > 840 |
| **non-parametric** |  |  |  |  |  |  |  |  |
| Steel–Dwass test |  |  |  |  |  |  | 0.0456 | 800 : 880 |

**zebrafish**

|  | Delay p-value | Delay wavelength | Duration p-value | Duration wave length | Angular velocity p-value | Angular velocity wavelength | Distance p-value | Distance wavelength |
|---|---|---|---|---|---|---|---|---|
| **parametric** |  |  |  |  |  |  |  |  |
| one-way repeated-measures ANOVA ($p$) |  |  | 0.0002 |  | 0.0010 |  |  |  |
| Shaffer's modified sequentially rejective |  |  | 0.0091 | 780 > 880 | 0.0031 | 780 > 820 |  |  |
| Bonferroni procedure (adjusted $p$) |  |  | 0.0182 | 780 > 840 | 0.0043 | 780 > 880 |  |  |
| **non-parametric** |  |  |  |  |  |  |  |  |
| Steel–Dwass test | n.s. |  |  |  |  |  | 0.0456 | 780 : 820 |
|  |  |  |  |  |  |  | 0.0456 | 780 : 840 |
|  |  |  |  |  |  |  | 0.0456 | 780 : 860 |
|  |  |  |  |  |  |  | 0.0456 | 780 : 880 |
|  |  |  |  |  |  |  | 0.0456 | 800 : 880 |
|  |  |  |  |  |  |  | 0.0456 | 840 : 880 |

**guppy**

|  | Delay p-value | Duration p-value | Duration wavelength | Angular velocity p-value | Angular velocity wavelength | Distance p-value | Distance wavelength |
|---|---|---|---|---|---|---|---|
| **parametric** |  |  |  |  |  |  |  |
| one-way repeated-measures ANOVA ($p$) | 0.2540 | 0.0248 |  | 0.1050 |  |  |  |
| Shaffer's modified sequentially rejective | — | n.s. |  | n.s. |  |  |  |
| Bonferroni procedure (adjusted $p$) |  |  |  |  |  |  |  |

**stickleback**

|  | Delay p-value | Duration p-value | Angular velocity p-value | Angular velocity wavelength | Distance p-value | Distance wavelength |
|---|---|---|---|---|---|---|
| **parametric** |  |  |  |  |  |  |
| one-way repeated-measures ANOVA ($p$) | 0.0008 | 0.0002 | 0.0001 |  | 0.0000 |  |
| Shaffer's modified sequentially rejective | n.s. | n.s. | 0.0030 | 780 > 860 | 0.0139 | 780 > 880 |
| Bonferroni procedure (adjusted $p$) |  |  | 0.0126 | 780 > 880 | 0.0139 | 780 > 820 |
|  |  |  |  |  | 0.0159 | 800 > 840 |
|  |  |  |  |  | 0.0357 | 800 > 860 |
|  |  |  |  |  | 0.0415 | 780 > 860 |
|  |  |  |  |  | 0.0467 | 780 > 840 |
| **non-parametric** |  |  |  |  |  |  |
| Steel–Dwass test |  |  |  |  | n.s. |  |

**mbuna**

|  | Delay p-value | Duration p-value | Angular velocity p-value | Distance p-value |
|---|---|---|---|---|
| **parametric** |  |  |  |  |
| one-way repeated-measures ANOVA ($p$) | 0.6728 | 0.2685 | 0.0194 | 0.8904 |
| Shaffer's modified sequentially rejective | — | — | n.s. | — |
| Bonferroni procedure (adjusted $p$) |  |  |  |  |

most sensitive parameter in zebrafish was Distance, and measurements at 840 nm yielded a *p*-value of 0.0456 (840 > 880, *t* = 2.8823). Therefore, in zebrafish, the threshold wavelength was around 840 nm.

### 3.4.3. Stickleback

Statistics of Delay and Duration gave *p*-values under 0.05 (one-way repeated-measures ANOVA, Delay: $F_{5,25} = 6.1611$, $p = 0.0008$; Duration: $F_{5,25} = 7.7186$, $p = 0.0002$). The parameter Distance was most sensitive in this fish, and the value at 800 nm significantly differed from those of longer wavelengths (800 > 840, *t* = 6.2013, *p* = 0.0159). Distance at 800 nm did not differ from the value at 820 nm. Stickleback lost their vision around 820 nm.

### 3.4.4. Mbuna

All four parameters did not change under light from 780 to 880 nm. In second-round OMR assay, mbuna did not perceive any of the light tested. When we consider the first- and second-round results together, the threshold wavelength for this species must be between 750 and 780 nm.

### 3.4.5. Goldfish

The statistics for the four parameters yielded $p < 0.05$. However, the value of Distance at 880 nm was close to zero but still positive, so we postulated that goldfish could perceive light around 860–880 nm. To verify this hypothesis, the data of 780, 800, 820, 840, 860 and 880 nm were subjected to Steel's test with the behavioural data of 950 nm as standard. Delay, Duration and Angular velocity at 880 nm gave significant *p*-values: 0.029 for Delay (*t* = 2.7654), 0.027 for Duration (*t* = 2.7851) and 0.0021 for Angular velocity (*t* = 3.5549). Distance at 860 nm differed from that at 950 nm (*t* = 2.5774, *p* = 0.048). The visible–invisible boundary of goldfish was between 860 and 880 nm.

### 3.4.6. Guppy

All four parameters increased or decreased, but none changed significantly through all the wavelengths tested. However, as Distance decreased to a negative value between 820 and 840 nm, guppy may have lost vision around these wavelengths.

## 4. Discussion

### 4.1. Our OMR strategy is useful for assessing the behaviour of many fish species

In this study, we compared red-light sensitivity among fish species. Our OMR strategy was established using medaka as a model [25]. We found that the strategy is useful for other fish species as well as medaka. Moreover, from the viewpoint of animal ethics, our procedure is an ethical way to examine the vision of animals. We assessed animal behaviour without administering drugs to the fish or restraining their movements. In this study, we showed that the procedure distinguished OMR-positive and OMR-negative results in several fish (figures 3–5). Our OMR test did not recognize non-OMR behaviour as being OMR positive as demonstrated by the results with Mexican cavefish (figure 3).

The best speed of stimulus, bandwidth of stripe and size of apparatus to induce an OMR might vary by fish species. In our previous studies [24,25,45], the drum was rotated at 6 r.p.m. and increased to 10 r.p.m. in this study. At 6 r.p.m., fish sometimes stopped following the stimulus. When the rotation speed was increased, fish focused on following the stripes when they were OMR positive but not if they were OMR negative. It was unclear whether 30 s of acclimatization was best for fish to resume normal behaviour. It is possible that the longer the acclimatization time, the better the fish response. We used light with different intensity for each wavelength (figure 2). Much more accurate information could be derived from determining the threshold intensity that evokes a response at each wavelength using a variety of intensities of light. Furthermore, it is necessary to verify the optimum conditions for inducing the OMR for each fish species such as stripe width, distance from stripes and rotation speed of striped pattern. In this study, a significant difference between species was shown under consistent conditions. However, the sensitivity of fish OMR measured under various experimental conditions as described above may be different from the results of this study.

Because Japanese striped loach stays still in nature, performing the behavioural test was not easy. Changing the speed of the rotation or stripe pattern could solve this problem. Senegal bichir, which also seldom swam, moved their head towards the rotating direction (electronic supplementary material, video file S2). Additional parameters that focus on such slight movement of an animal could help solve this issue. Other strategies, such as OKR and ERG, are good ways to test the vision of an animal and could be appropriate to check the visual sensitivity of these two species. It is important to note that these two behavioural tests involve restricting animals, which may present an ethical issue.

For fish of larger size, such as adult rainbow trout, Senegal bichir and cichlids (Nile tilapia and mbuna), a large testing apparatus may be helpful. Our spectrograph horizontally dispersed a series of monochromatic lights of wavelength 250–1000 nm on a U-shaped focal surface of about 10 m wide. Hence, a bigger apparatus and a larger aquarium covered a wider wavelength range. Behavioural assays under the wider range of wavelength would result in a larger statistical dispersion. From this standpoint, juvenile fish were used in this work. Development is known to affect vision. For example, salmon changes its A1/A2 chromophore seasonally [46], and the shift of the A1/A2 ratio correlates with the fish's life cycle [47]. The transition from A1 to A2 results in a red shift of the visual pigment's absorption spectrum [14,48]. As well as a change of chromophore usage, fish express a different subset of opsins at different developmental stages [4,49–51]. In single cones of salmon, the youngest fish (yolk-sac alevins) express opsin SWS1, whereas older juveniles (smolts) express SWS2 predominantly. In this way, opsin and its chromophore usage change, depending on fish ontogeny. Accordingly, sensitivities to deep-red light in adult fish may possibly differ from our current results.

All four parameters helped to distinguish OMR behaviour from non-OMR behaviour. When an animal gradually lost its sight, Delay increased, and Duration, Angular velocity and Distance decreased. Angular velocity showed little change in slow swimmers such as goldfish (figure 5). In the first-round of OMR tests under light of wavelength 700–1000 nm, Distance of all six fish species converged around zero. Moreover, in the second-round OMR test, according to the estimation of a visible–invisible boundary in medaka, stickleback and zebrafish, Distance also dropped to around zero. This suggests that by the point at which Distance was close to zero, the animals had lost their vision.

## 4.2. The four parameters unveiled the red-light sensitivity of fish

In past reports, zebrafish and guppy have shown phototaxis under light of wavelength greater than 900 nm [37], but they did not respond to light greater than 800 nm in the OMR assay [29,31]. Our OMR strategy detected OMR behaviour at a longer wavelength than previously reported (electronic supplementary material, table S1). Goldfish were OMR positive at 860 nm. The reason why goldfish could perceive light of longer wavelength in this study is not clear. LWS with A2 chromophore is a candidate to accomplish the perception of red light (figure 1). Goldfish have developed from ancient Asian carp, and carp prefers still water and tends to hide in mud and water grass. In such circumstance, high sensitivity to deep-red light could be a survival advantage [37]. Like goldfish, Nile tilapia live in murky water. In this work, based on four OMR parameters, tilapia could perceive light of wavelength 720–860 nm.

## 4.3. The wavelengths of light that invoked optomotor response correlated with expressed opsin repertoire

In this study, we estimated the longest wavelength that provoked an OMR (figure 1b). Besides mbuna, Nile tilapia also belongs to the family Cichlidae, but their habitats are different. Nile tilapia inhabit shallow waters with increased turbidity. Mbuna are a brightly coloured rock dweller. The opsin repertoires of these two fish were not the same (figure 1b). Mbuna and Nile tilapia varied in the expression of opsin genes. In Nile tilapia, *LWS* gene expression covered about 80% of overall opsin expression. Mbuna expressed *RH2* and *SWS2-B* genes predominantly, followed by *SWS1* and *LWS* genes [15]. As referenced in the study of Sabbah *et al.* [52], mbuna do not possess *LWS*, whereas Carleton & Kocher [15] found that *LWS* in mbuna constituted less than 10% of all opsin gene expression. According to Carleton *et al.* [53], mbuna appear to possess UV-shifted visual pigments. We showed that Nile tilapia could perceive deep-red light, and mbuna did not detect wavelengths over 780 nm. The expressed opsin repertoire could explain the OMR sensitivity to NIR light.

In goldfish, LWS-expressing cones elicit an OMR at wavelengths between 409 and 699 nm, while in adult zebrafish, an OMR occurs between 416 and 699 nm [30,31]. In zebrafish larvae on day 7 post-fertilization, cones with LWS or RH2 detect the stimuli [54]. That low *LWS* opsin levels in adult

zebrafish [13] provoked an OMR is quite interesting. Although the expressed *LWS* opsin was present in a slight amount overall in the retina, this low amount appears to be enough to cause a behavioural response in zebrafish. According to Zimmermann *et al*. [55], different parts of the retina are assigned to separate roles in colour vision, achromatic vision, scotopic vision and UV vision of prey capture. If this is the case in zebrafish, *LWS* opsin might be expressed in a certain part of the retina in sufficient concentration that allows the detection of moving stimuli.

We found that the number of opsin genes did not correlate with red-light sensitivity of fish (figure 1; electronic supplementary material, table S1). Two *LWS* opsin genes are expressed in medaka and zebrafish, four in guppy and one each in the other three fish species. The longest wavelength that elicited an OMR was 860 nm in goldfish; in zebrafish, 840 nm; in medaka and guppy, 820 nm; in stickleback, 800 nm and in mbuna, between 750 and 780 nm. Evolutionary duplications of *LWS* opsin genes may have improved other ophthalmic functions, such as the ability to discriminate in certain wavelength ranges. On the other hand, in fish, opsin expression and activity change during the day. It is possible that differences in expression levels due to circadian rhythms, rather than *LWS* opsin copy number, have had some influence on our results [56,57].

The longest wavelength that provoked an OMR in medaka, goldfish, zebrafish, guppy and stickleback was greater than or equal to 800 nm, while being under 780 nm in mbuna. In medaka, the absorption spectrum of LWS pigment was almost zero at 650 nm [9]; in stickleback, 650 nm [10]; in Nile tilapia, 675 nm [11] and in guppy, 650 nm [12]. Paradoxically, LWS pigment of these fish could absorb light of longer wavelength. The absorption spectra of visual pigment in these fish species were measured with retinal A1 as a chromophore. In zebrafish and goldfish, retinal A2 works as a chromophore *in vivo* [14,48]. With retinal A2, visual pigment absorbs light of longer wavelength, with shifts of $\lambda_{max}$ of about 10–60 nm. In goldfish, $\lambda_{max}$ of LWS with A1 is 566 nm, while that with A2 is 618 nm [14]. In zebrafish, $\lambda_{max}$ of LWS with A2 is 618 nm and is red-shifted 57 nm longer than that with A1 [48]. Retinal A2 red-shifted not only the $\lambda_{max}$ but pushed up all the absorption spectra. With retinal A2, in goldfish, absorption spectra of LWS pigment almost reached 750 nm [14], and 800 nm in zebrafish [48]. The gene *Cyp27c1*, which has a role in switching from A1 to A2 chromophore usage in zebrafish [48], is also present in the guppy genome. However, adult guppy use A1, not A2, as a chromophore [58].

When we considered the use of retinal A1 or A2 as a chromophore, the absorption spectra of visual pigments appeared to correlate with the wavelengths that invoked an OMR (figure 1*b*). Even so, a gap still exists between them. Some mechanisms achieve the OMR in deep-red and NIR light. Rhodopsin could play a role in the perception of longer wavelength light. In our previous studies using light-adapted medaka, we found that LWS, not rhodopsin, activated the OMR [24]. However, it was unclear whether rhodopsin similarly did not activate an OMR under NIR light in other fish species after light adaptation. Non-cone, non-rod visual pigments may be involved in the perception of red light. For example, in humans, melanopsin is involved in the perception of brightness and colour [59,60]. In teleost, such as zebrafish and the African cichlid (*Astatotilapia burtoni*), melanopsin was identified [61–64]. However, as melanopsin is maximally sensitive at around 480 nm [62,65], it is very unlikely that melanopsin mediates long-wavelength sensitivity. Other non-cone, non-rod visual pigments are also candidates in red-light perception [66–71]. Alternatively, the gap may reflect photoreceptors' ability to respond over a range on many log units. If visual pigments are displayed using log absorbance, they could easily mediate longer wavelength sensitivity. Future studies testing the OMR using fish with knock-out LWS, rhodopsin genes or non-cone, non-rod visual pigments component genes will solve this question.

Ethics. The experiments were conducted in accordance with the Guidelines for Proper Conduct of Animal Experiments and approved by the National Institute for Basic Biology, Aichi, Japan (18A093).

Data accessibility. The behavioural data related to this article are available within the Dryad Digital Repository: https://doi.org/10.5061/dryad.jh9w0vt88 [72].

Authors' contributions. S.F. conceived and supervised this study. Y.K. technically assisted in OMR assays and M.M. and S.F. conducted OMR assays. M.M. analysed fish behaviour. M.M. wrote, and all the authors edited and approved the manuscript.

Competing interests. The authors have no competing interests to declare.

Funding. M.M. was funded by Tsuji Foundation of Japan Women's University (year of 2018). This research was supported by a grant of Joint Research by the National Institutes of Natural Sciences (NINS; no. 01111904), a Grant-in-Aid for Scientific Research (C) from the Japan Society for the Promotion of Science (JSPS; no. 17K07506) and National Institute for Basic Biology's Collaborative Research Program (18-101, 19-101) to S.F.

Acknowledgements. We thank Mizuki Sano, Yuko Ogawa, Mayu Sato, Risako Masuda at JWU, and Joe Sakamoto, Maki Kondo at NIBB for assisting in the OMR test. We also thank Maki Kondo at NIBB for measuring all the spectra of light used in this work.

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
