## [Peer Review File · Royal Society Open Science]

Review History

RSOS-201005.R0 (Original submission)

Review form: Reviewer 1

Is the manuscript scientifically sound in its present form?

No

Are the interpretations and conclusions justified by the results?

Yes

Is the language acceptable?

Yes

Do you have any ethical concerns with this paper?

No

Have you any concerns about statistical analyses in this paper?

No

Recommendation?

Major revision is needed (please make suggestions in comments)

Comments to the Author(s)

The manuscript entitled "A comparison of behavioral red-light sensitivity in fish according to the optomotor response" by Matsuo et al. uses an optomotor response (OMR) assay previously developed in Matsuo et al. (2018) *Biology Open* 7:bio033175 to measure far-red visual sensitivity of thirteen species of teleost fish. They found evidence of OMR in eight of thirteen species with further analyses on six species, namely, zebrafish, guppy, medaka, goldfish, stickleback, mbuna. They describe the longest wavelength to elicit OMR behaviour in those six species and compare those data to previously reported maxima measured by various other assays. By all indications the OMR assay is sound and the data are a valuable contribution to the field. However, the manuscript is not well written and makes interpreting/confirming the results difficult. I appreciate the effort that went into husbandry and data analysis and I think with extensive edits and the addition of modest amounts of data the manuscript is a good fit in Royal Society Open Science. There are a number a technical concerns that I would like to see addressed and I have offered extensive, albeit not comprehensive, copy-edit suggestions to aid the authors as they rewrite/edit the manuscript to improve its clarity.

Concerns/suggestions follow in a chronological order for each header, with page and line numbers (p#l#, page number and line number, respectively) indicated where applicable:

Technical concerns:

P4L32-50: Is the origin of each species known? Are they wild-caught or hatchery/lab reared?

Also, stickleback referred to Broughton Archipelago, is this appropriate? We can't know with the information provided.

P4L32: What is the ambient light source used? Strong evidence of lights impacting opsin expression (e.g., Fuller & Claricoates 2011 *Molecular Ecology* 20, 3321-3335) may influence the assay.

P4L58: How long were fish light adapted for prior to the behavioural assay? Did you control for circadian rhythms? There is evidence that both opsin expression and activity vary with time of day so this may be of concern for your OMR assay.

P4L59-60: What are the spectra for the different wavelengths from the monochromator? I notice many of the wavelengths are similar to those described in previous work (i.e., Matsuo et al. 2018) but the photon densities differ. Please provide spectra for all wavelengths used in the assay. A multi-axis figure including spectra of ambient light and monochromator light would be useful.

P5L5: Fish were acclimated to the assay for 30 seconds, although it is not clear if they were transferred to a new behavioral arena or if the OMR rig was lowered into the aquaria. If they were physically removed, e.g., by hand net, then 30 seconds does not seem adequate for the fish to recover equilibria and resume normal behaviours. Please elaborate.

P6L9-14: Can you elaborate on why you excluded tetra and corydoras from later analyses? Here you describe them as OMR positive and refer to video files but there is no follow-up.

P6L31-41: Please provide statistics/p-values supporting the statements herein.

P7L25: p-values are used but please report the F-statistic or any other appropriate statistic for the parametric/non-parametric tests used.

P8L31: What about stickleback 800>820 comparison?

P8L49-51: Are you comparing data from the first round and second round? Please clarify.

P9L59-P10L8: The speed of the OMR apparatus is clearly important. Does this need to be empirically calibrated per species, or perhaps per individual based on size?

P10L34: reference to "not fully-grown fish", read: juvenile. Ontogeny is known to affect vision. Some description of the limitations of the assay for larger fish and the stage of development of fishes used in the present study is warranted. Same goes for retinal and dehydroretinal and its impact on visual pigments. Some species use both vitamins A1 and A2 depending on season, life history. The manuscript briefly described A1/A2 but a more detailed description of this phenomenon is warranted.

Spec of opsins from table 1, figure 2... maxima only tells us so much, what is half-bandwidth, or better yet, provide the full spectra often reported in the studies you refer to. Many assays (e.g., microspectrophotometry) do not include infrared in their recordings but some may? What about non-visual opsins playing a role in sensitivity. Melanopsin contributes to involuntary pupillary reflex in humans.

Tables 3 & 4: If displayed in a table like this, please fill out all the appropriate statistics regardless of significance. Or provide these data in a supplementary table. Bold or highlight those that are significant if you want to emphasize those data. Perhaps bold/highlight the greatest difference/threshold data you describe in text.

Please add the data from cavefish on the plots from Fig 3 to have a ready comparison on an OMR negative fish.

Figure 5 & 6: What do the broken lines indicate? Also, why were guppy and medaka not tested at 900 and 1000 nm? The exclusion of these without explanation is concerning.

Figure 7: Data are depicted for a medaka LWS knock-out, where were these data derived from? The red box around medaka LWS, how did you arrive at this box, was it empirically tested? The lack of transparency here is concerning.

The statement that the maximum wavelength eliciting OMR correlates with LWS spectra is tenuous at best. That claim is made in the abstract as if it were empirically tested but that is not the case.

Copy-edit, clarity, and other minor comments:

P2L39: "the visibility" is awkward, alternatives: visual ability, vision, visual sensitivity

P2L44-45: "...sake of no or reckless swimming" consider rewording. I am not sure what you mean by "reckless swimming"

P2L51: "absorption spectra" these data are not shown, you only describe max sensitivity.

P2L53: multiplication should read duplication

P2L54: "might aid not increasing" grammar, reword

P3L14: The universal paradigm of one rod opsin may be in question, see: Musilova et al. Science. 2019; 364(6440): 588-592.

P3L15-16: "...every cone includes several type of cone visual pigments..." as written this statement incorrectly implies opsin co-expression occurs in every cell.

P3L31: "...the spectral sensitivity..." omit "the"

P3L37: "The schooling behavior..." implies you've introduced schooling behavior as a concept prior. Omit "the" or reword the sentence. "This Mexican cavefish" omit "this"

P3L31-42: Consider rewording this entire section for clarity. The last two sentences are particularly difficult to understand.

P3L45: Consider writing about how the diversity of opsins is driven mostly by tandem duplication.

P3L60: "Besides the OMR, fish vision was examined..." as written this sentence suggests the present study used ERG, phototaxis, OKR. Reword.

P4L4: "...in guppy as longest", as longest what? Unclear.

P4L42: "till" should read "until" this misspelling occurs in multiple instances throughout the manuscript.

P4L49: How many animals were used for each species? Why did you use fewer animals for these species? Provide justification.

P5L10: "...allocated ID..." is a strange way to phrase this. Is the purpose of this statement to imply the person conducting downstream analyses was "blind" e.g., unaware of the species they were analyzing to reduce bias?

P5L14-17: "...exclude the noise. Guppy male with..." a little awkward, consider rewording e.g., "...exclude the noise, for example, tracking was made difficult when a male guppy's long tail fin disturbed the water surface."

P5L23-31: The use of parentheses here is unnecessary. Also, the description of distance as "plus" and "minus" is confusing. Suggestion: "4. Distance: the overall swimming distance a fish swims

in the direction of the rotating apparatus. When a fish swims in the opposite direction that distance is subtracted from the overall distance for the trial.”

P5L30: misplaced “

P6L5: Header, “applicable” and “inapplicable” is awkward. Use instead included/excluded? Or initial OMR test, and you found a number that were OMR negative.

P6L28: NIR is not defined in-text

P6L48: “These fish were further testes their behavior.” testes should be tested, also the grammar of this sentence is incorrect. The first three paragraphs of 3.3 should be restructured for better grammar/clarity.

P6L56-P7L8: “soared” should be “increased”, “went down” should be “decreased”. “Steep drop in the graphs” is a strange way of describing the data, consider rewording. This applies to later examples throughout the text where you refer to the “graphs of...” or “the line graph of...”

P7L5: “animal” reword, instead “an animal” or “fish”

P7L15-28: this is the first mention of a “ratio” in text. What is this referring to? Describe in methods first.

P7L31: the speed of the apparatus belongs in the methods.

P7L42: introducing a new metric, rounds/test how is this different from distance? Describe in more detail in the methods section.

You ran an initial test for OMR and then a subsequent test for NIR sensitivity using the OMR assay. That could be clarified in the methods section.

P8L26: Report the p-values and F-statistics.

P8L42: “(840:880)” format differs from above. Be consistent.

P8L55-P9L5: This section may be more appropriate in the discussion.

P9L3: “almost touched to horizontal axis” reword, describe the data, not the line/graph itself.

P9L26: “could lost” reword

P9L40-41: “Moreover, ...” This sentence is confusing, what are you trying to say here?

P9L54: What is scrambled and loitered as it relates to fish behaviour? Please reword/clarify.

P10L11: Reword this first sentence.

P10L14: “...affected by chance...” was it or was it not?

P10L17-18: Was distance the most useful tool? You described all four as useful above. This is confusing.

P10L20-21: “The number over zero...” I do not understand what this sentence is trying to convey.

P10L29: “not fully-grown fish” e.g., juvenile?

P11L26 and P11L45: “et al” should read “et al.” al. is the abbreviated form of alia

P11L28: what is meant here by “extend”?

The format of figure captions is inconsistent.

Figures 1 and 2 could be combined for a single visually appealing figure.

Figure 2 and 7 appear to convey similar information, perhaps combine them. A multi-panel figure with phylogeny (as mentioned above) would be a great summary figure.

Figures 3-6 need better labelling and provide more detailed figure caption for Figs.3 and 4

Review form: Reviewer 2

Is the manuscript scientifically sound in its present form?

No

Are the interpretations and conclusions justified by the results?

No

Is the language acceptable?

No

Do you have any ethical concerns with this paper?

No

Have you any concerns about statistical analyses in this paper?

No

Recommendation?

Reject

Comments to the Author(s)

There have been many studies of spectral sensitivity in teleost fish using a variety of methods. One of the strengths and weaknesses of such research is that different methods give different results and it is therefore often hard to compare data derived from different studies. Thus, a comparison of several species, such as is attempted here, using the same method for each species could be interesting.

The authors have chosen to only look at long wavelengths (>720nm). Why they have done this is not explained. Most studies of spectral sensitivity include the whole of the 'visible' spectrum. It must be explained early on why this was not done.

This study also claims to look at 'sensitivity' at long wavelengths. What the authors did was to determine the response of fish to a set intensity for each wavelength. Much more accurate information could be derived from determining the threshold intensity that evokes a response at each wavelength using a variety of different intensities.

The first paragraph of the introduction sets out basic visual concepts. Unfortunately, it is possible to take issue with many statements.

- "There are two types of photoreceptor....." This is only true when considering the outer retina. However, the inner retina contains distinct photoreceptive cells (eg melanopsin)
- "Rods contain a single visual pigment" This is not true for amphibia or many deep-sea fish.
- "Integration of (cone) signals enable animals to discriminate colour...." Not if the 'integration is simply additive.

I suggest this paragraph be adjusted to be less generalised.

Introduction para 2

- *Astyanax Mexicanus* specific names should start with a lower case letter
- Why is the scientific name for the bichir not given, when that of the cave fish is?

Introduction para 3 this seems to largely repeat table 1. Is this necessary?

Introduction, para 4 "...fish vision was examined in several ways ...(Table 1)..." Initially I thought this was referring to this study, but it is clear it is referring to previous literature. Although some studies are listed I know it is not an exhaustive list. Salmonid spectral sensitivity, for example, has been widely studied and some rainbow trout studies are not listed. The same is true for goldfish and perhaps other species listed. Is what is listed the study with the longest red sensitivity? If so, this should be clarified.

The legend of Figure 5 fully explains the figure. However, the legend of the preceding 2 figures (Figures 3&4) gives no such information, limiting their value. In fact, as Figures 3&4 are negative (i.e. show there was no optomotor response). I wonder whether they are needed in the main text at all.

Many of the numerical results given in the text are repetitions of data in tables. This seems unnecessary.

While the data in Figure 5 seems clear, trends in Figure 6 are much less easy to make out.

Figure 7 it is unclear what the blue smudge indicates. The legend says it is the wavelengths which elicited an OMR. However, this cannot be right as shorter wavelengths will have elicited an OMR too. Is it the longest wavelength that elicited an OMR? Why is it a gradient; surely it should be a single value?

Decision letter (RSOS-201005.R0)

Dear Dr Matsuo

The Editors assigned to your paper RSOS-201005 "A comparison of behavioral red-light sensitivity in fish according to the optomotor response" have made a decision based on their reading of the paper and any comments received from reviewers.

Regrettably, in view of the reports received, the manuscript has been rejected in its current form. However, a new manuscript may be submitted which takes into consideration these comments.

We invite you to respond to the comments supplied below and prepare a resubmission of your manuscript. Below the referees' and Editors' comments (where applicable) we provide additional requirements. We provide guidance below to help you prepare your revision.

Please note that resubmitting your manuscript does not guarantee eventual acceptance, and we do not generally allow multiple rounds of revision and resubmission, so we urge you to make every effort to fully address all of the comments at this stage. If deemed necessary by the Editors, your manuscript will be sent back to one or more of the original reviewers for assessment. If the original reviewers are not available, we may invite new reviewers.

Please resubmit your revised manuscript and required files (see below) no later than 16-Mar-2021. Note: the ScholarOne system will 'lock' if resubmission is attempted on or after this deadline. If you do not think you will be able to meet this deadline, please contact the editorial office immediately.

Please note article processing charges apply to papers accepted for publication in Royal Society Open Science (<https://royalsocietypublishing.org/rsos/charges>). Charges will also apply to papers transferred to the journal from other Royal Society Publishing journals, as well as papers submitted as part of our collaboration with the Royal Society of Chemistry (<https://royalsocietypublishing.org/rsos/chemistry>). Fee waivers are available but must be requested when you submit your manuscript (<https://royalsocietypublishing.org/rsos/waivers>).

Thank you for submitting your manuscript to Royal Society Open Science and we look forward to receiving your resubmission. If you have any questions at all, please do not hesitate to get in touch.

on behalf of Prof Kevin Padian (Subject Editor)
openscience@royalsociety.org

Associate Editor Comments to Author:

Given the critiques of the referees, we do not feel that the timeline of a major revision would be sufficient for you to reasonably respond. With this in mind, we are going to reject the current iteration of your paper, but invite you to resubmit a thoroughly revised paper (including a substantial linguistic edit - though we sympathise that English is a peculiar and inconsistent language), which will receive further review.

Editor comments to author:

We now have two detailed and constructive reviews of your manuscript. Based on these, I agree with the AE's comments and ask that you take time to consider the reviews and revise your manuscript accordingly. Additionally, the English will need to be edited by someone who is a native Anglophone with experience in your field, or we will unfortunately be unable to publish it. Please accept our regrets that English is such a difficult and irregular language; still, the work must be comprehensible to readers. A reject/resub decision gives you more time to revise than our 3-week "major revision" decision. We look forward to a resubmission, but please note that it must address successfully the comments of the reviewers and be in publishable English. Best wishes.

Reviewer comments to Author:

Reviewer: 1

Comments to the Author(s)

The manuscript entitled "A comparison of behavioral red-light sensitivity in fish according to the optomotor response" by Matsuo et al. uses an optomotor response (OMR) assay previously developed in Matsuo et al. (2018) *Biology Open* 7:bio033175 to measure far-red visual sensitivity of thirteen species of teleost fish. They found evidence of OMR in eight of thirteen species with further analyses on six species, namely, zebrafish, guppy, medaka, goldfish, stickleback, mbuna. They describe the longest wavelength to elicit OMR behaviour in those six species and compare those data to previously reported maxima measured by various other assays. By all indications the OMR assay is sound and the data are a valuable contribution to the field. However, the manuscript is not well written and makes interpreting/confirming the results difficult. I appreciate the effort that went into husbandry and data analysis and I think with extensive edits and the addition of modest amounts of data the manuscript is a good fit in Royal Society Open Science. There are a number of technical concerns that I would like to see addressed and I have offered extensive, albeit not comprehensive, copy-edit suggestions to aid the authors as they rewrite/edit the manuscript to improve its clarity.

Concerns/suggestions follow in a chronological order for each header, with page and line numbers (p#l#, page number and line number, respectively) indicated where applicable:

Technical concerns:

P4L32-50: Is the origin of each species known? Are they wild-caught or hatchery/lab reared? Also, stickleback referred to Broughton Archipelago, is this appropriate? We can't know with the information provided.

P4L32: What is the ambient light source used? Strong evidence of lights impacting opsin expression (e.g., Fuller & Claricoates 2011 *Molecular Ecology* 20, 3321-3335) may influence the assay.

P4L58: How long were fish light adapted for prior to the behavioural assay? Did you control for circadian rhythms? There is evidence that both opsin expression and activity vary with time of day so this may be of concern for your OMR assay.

P4L59-60: What are the spectra for the different wavelengths from the monochromator? I notice many of the wavelengths are similar to those described in previous work (i.e., Matsou et al. 2018) but the photon densities differ. Please provide spectra for all wavelengths used in the assay. A multi-axis figure including spectra of ambient light and monochromator light would be useful.

P5L5: Fish were acclimated to the assay for 30 seconds, although it is not clear if they were transferred to a new behavioral arena or if the OMR rig was lowered into the aquaria. If they were physically removed, e.g., by hand net, then 30 seconds does not seem adequate for the fish to recover equilibria and resume normal behaviours. Please elaborate.

P6L9-14: Can you elaborate on why you excluded tetra and corydoras from later analyses? Here you describe them as OMR positive and refer to video files but there is no follow-up.

P6L31-41: Please provide statistics/p-values supporting the statements herein.

P7L25: p-values are used but please report the F-statistic or any other appropriate statistic for the parametric/non-parametric tests used.

P8L31: What about stickleback 800>820 comparison?

P8L49-51: Are you comparing data from the first round and second round? Please clarify.

P9L59-P10L8: The speed of the OMR apparatus is clearly important. Does this need to be empirically calibrated per species, or perhaps per individual based on size?

P10L34: reference to "not fully-grown fish", read: juvenile. Ontogeny is known to affect vision. Some description of the limitations of the assay for larger fish and the stage of development of fishes used in the present study is warranted. Same goes for retinal and dehydroretinal and its impact on visual pigments. Some species use both vitamins A1 and A2 depending on season, life history. The manuscript briefly described A1/A2 but a more detailed description of this phenomenon is warranted.

Spec of opsins from table 1, figure 2... maxima only tells us so much, what is half-bandwidth, or better yet, provide the full spectra often reported in the studies you refer to. Many assays (e.g., microspectrophotometry) do not include infrared in their recordings but some may? What about non-visual opsins playing a role in sensitivity. Melanopsin contributes to involuntary pupillary reflex in humans.

Tables 3 & 4: If displayed in a table like this, please fill out all the appropriate statistics regardless of significance. Or provide these data in a supplementary table. Bold or highlight those that are significant if you want to emphasize those data. Perhaps bold/highlight the greatest difference/threshold data you describe in text.

Please add the data from cavefish on the plots from Fig 3 to have a ready comparison on an OMR negative fish.

Figure 5 & 6: What do the broken lines indicate? Also, why were guppy and medaka not tested at 900 and 1000 nm? The exclusion of these without explanation is concerning.

Figure 7: Data are depicted for a medaka LWS knock-out, where were these data derived from? The red box around medaka LWS, how did you arrive at this box, was it empirically tested? The lack of transparency here is concerning.

The statement that the maximum wavelength eliciting OMR correlates with LWS spectra is tenuous at best. That claim is made in the abstract as if it were empirically tested but that is not the case.

Copy-edit, clarity, and other minor comments:

P2L39: "the visibility" is awkward, alternatives: visual ability, vision, visual sensitivity

P2L44-45: "...sake of no or reckless swimming" consider rewording. I am not sure what you mean by "reckless swimming"

P2L51: "absorption spectra" these data are not shown, you only describe max sensitivity.

P2L53: multiplication should read duplication

P2L54: "might aid not increasing" grammar, reword

P3L14: The universal paradigm of one rod opsin may be in question, see: Musilova et al. Science. 2019; 364(6440): 588–592.

P3L15-16: "...every cone includes several type of cone visual pigments..." as written this statement incorrectly implies opsin co-expression occurs in every cell.

P3L31: "...the spectral sensitivity..." omit "the"

P3L37: "The schooling behavior..." implies you've introduced schooling behavior as a concept prior. Omit "the" or reword the sentence. "This Mexican cavefish" omit "this"

P3L31-42: Consider rewording this entire section for clarity. The last two sentences are particularly difficult to understand.

P3L45: Consider writing about how the diversity of opsins is driven mostly by tandem duplication.

P3L60: "Besides the OMR, fish vision was examined..." as written this sentence suggests the present study used ERG, phototaxis, OKR. Reword.

P4L4: "...in guppy as longest", as longest what? Unclear.

P4L42: "till" should read "until" this misspelling occurs in multiple instances throughout the manuscript.

P4L49: How many animals were used for each species? Why did you use fewer animals for these species? Provide justification.

P5L10: "...allocated ID..." is a strange way to phrase this. Is the purpose of this statement to imply the person conducting downstream analyses was "blind" e.g., unaware of the species they were analyzing to reduce bias?

P5L14-17: "...exclude the noise. Guppy male with..." a little awkward, consider rewording e.g., "...exclude the noise, for example, tracking was made difficult when a male guppy's long tail fin disturbed the water surface."

P5L23-31: The use of parentheses here is unnecessary. Also, the description of distance as "plus" and "minus" is confusing. Suggestion: "4. Distance: the overall swimming distance a fish swims in the direction of the rotating apparatus. When a fish swims in the opposite direction that distance is subtracted from the overall distance for the trial."

P5L30: misplaced "

P6L5: Header, "applicable" and "inapplicable" is awkward. Use instead included/excluded? Or initial OMR test, and you found a number that were OMR negative.

P6L28: NIR is not defined in-text

P6L48: "These fish were further testes their behavior." testes should be tested, also the grammar of this sentence is incorrect. The first three paragraphs of 3.3 should be restructured for better grammar/clarity.

P6L56-P7L8: "soared" should be "increased", "went down" should be "decreased". "Steep drop in the graphs" is a strange way of describing the data, consider rewording. This applies to later examples throughout the text where you refer to the "graphs of..." or "the line graph of..."

P7L5: "animal" reword, instead "an animal" or "fish"

P7L15-28: this is the first mention of a "ratio" in text. What is this referring to? Describe in methods first.

P7L31: the speed of the apparatus belongs in the methods.

P7L42: introducing a new metric, rounds/test how is this different from distance? Describe in more detail in the methods section.

You ran an initial test for OMR and then a subsequent test for NIR sensitivity using the OMR assay. That could be clarified in the methods section.

P8L26: Report the p-values and F-statistics.

P8L42: "(840:880)" format differs from above. Be consistent.

P8L55-P9L5: This section may be more appropriate in the discussion.

P9L3: "almost touched to horizontal axis" reword, describe the data, not the line/graph itself.

P9L26: "could lost" reword

P9L40-41: "Moreover, ..." This sentence is confusing, what are you trying to say here?

P9L54: What is scrambled and loitered as it relates to fish behaviour? Please reword/clarify.

P10L11: Reword this first sentence.

P10L14: "...affected by chance..." was it or was it not?

P10L17-18: Was distance the most useful tool? You described all four as useful above. This is confusing.

P10L20-21: "The number over zero..." I do not understand what this sentence is trying to convey.

P10L29: "not fully-grown fish" e.g., juvenile?

P11L26 and P11L45: "et al" should read "et al." al. is the abbreviated form of alia

P11L28: what is meant here by "extend"?

The format of figure captions is inconsistent.

Figures 1 and 2 could be combined for a single visually appealing figure.

Figure 2 and 7 appear to convey similar information, perhaps combine them. A multi-panel figure with phylogeny (as mentioned above) would be a great summary figure.

Figures 3-6 need better labelling and provide more detailed figure caption for Figs.3 and 4

Reviewer: 2

Comments to the Author(s)

There have been many studies of spectral sensitivity in teleost fish using a variety of methods. One of the strengths and weaknesses of such research is that different methods give different results and it is therefore often hard to compare data derived from different studies. Thus, a comparison of several species, such as is attempted here, using the same method for each species could be interesting.

The authors have chosen to only look at long wavelengths (>720nm). Why they have done this is not explained. Most studies of spectral sensitivity include the whole of the 'visible' spectrum. It must be explained early on why this was not done.

This study also claims to look at 'sensitivity' at long wavelengths. What the authors did was to determine the response of fish to a set intensity for each wavelength. Much more accurate information could be derived from determining the threshold intensity that evokes a response at each wavelength using a variety of different intensities.

The first paragraph of the introduction sets out basic visual concepts. Unfortunately, it is possible to take issue with many statements.

- "There are two types of photoreceptor....." This is only true when considering the outer retina. However, the inner retina contains distinct photoreceptive cells (eg melanopsin)
- "Rods contain a single visual pigment" This is not true for amphibia or many deep-sea fish.
- "Integration of (cone) signals enable animals to discriminate colour...." Not if the 'integration is simply additive.

I suggest this paragraph be adjusted to be less generalised.

Introduction para 2

- *Astyanax Mexicanus* specific names should start with a lower case letter
- Why is the scientific name for the bichir not given, when that of the cave fish is?

Introduction para 3 this seems to largely repeat table 1. Is this necessary?

Introduction, para 4 "...fish vision was examined in several ways ...(Table 1)..." Initially I thought this was referring to this study, but it is clear it is referring to previous literature.

Although some studies are listed I know it is not an exhaustive list. Salmonid spectral sensitivity, for example, has been widely studied and some rainbow trout studies are not listed. The same is true for goldfish and perhaps other species listed. Is what is listed the study with the longest red sensitivity? If so, this should be clarified.

The legend of Figure 5 fully explains the figure. However, the legend of the preceding 2 figures (Figures 3&4) gives no such information, limiting their value. In fact, as Figures 3&4 are negative (i.e. show there was no optomotor response). I wonder whether they are needed in the main text at all.

Many of the numerical results given in the text are repetitions of data in tables. This seems unnecessary.

While the data in Figure 5 seems clear, trends in Figure 6 are much less easy to make out.

Figure 7 it is unclear what the blue smudge indicates. The legend say it is the wavelengths which elicited an OMR. However, this cannot be right at shorter wavelengths will have elicited an OMR too. Is it the longest wavelength that elicited an OMR? Why is it a gradient; surely it should be a single value?

===PREPARING YOUR MANUSCRIPT===

===PREPARING YOUR REVISION IN SCHOLARONE===

Author's Response to Decision Letter for (RSOS-201005.R0)

See Appendix A.

RSOS-210415.R0

Review form: Reviewer 1

Is the manuscript scientifically sound in its present form?

Yes

Are the interpretations and conclusions justified by the results?

Yes

Is the language acceptable?

Yes

Do you have any ethical concerns with this paper?

No

Have you any concerns about statistical analyses in this paper?

No

Recommendation?

Accept as is

Comments to the Author(s)

The revised manuscript by Matsuo et al. entitled "Behavioural red-light sensitivity in fish according to the optomotor response" is a valuable addition to the field and in its current state is suitable for publication. The authors conscientiously addressed my critical and rather lengthy comments from the first round of review and I appreciate their efforts. I particularly appreciate the lengths the authors went clarifying the methods and making them more reproducible, the added sections on ontogeny and chromophores added interesting context, and the figures are much more informative. Figure one in particular is a marked improvement and conveys a lot of really useful details -- my one comment would be to include the data reported here in Fig 1, not just previously reported behavioural and ERG data from the literature (e.g., goldfish are shown to have OMR from wavelengths approximately <650nm but the authors report much longer wavelengths eliciting an OMR response. Overall, I think these data are reported well in the manuscript and publication in the Royal Society Open Science is justified.

Review form: Reviewer 2

Is the manuscript scientifically sound in its present form?

Yes

Are the interpretations and conclusions justified by the results?

No

Is the language acceptable?

Yes

Do you have any ethical concerns with this paper?

No

Have you any concerns about statistical analyses in this paper?

No

Recommendation?

Reject

Comments to the Author(s)

In my original review I said "The authors have chosen to only look at long wavelengths (>720nm). Why they have done this is not explained. Most studies of spectral sensitivity include the whole of the 'visible' spectrum. It must be explained early on why this was not done." In their response letter the authors state that this was due to a "technical issue". Specifically, it seems to concern the well-known phenomenon of fluorescence in the UV. This to me still does not explain why wavelengths between the far red and the UV were not examined. There may well be a good reason for only looking at longer wavelengths, but as I said previously 'It must be explained early on (i.e in the Introduction) why this (examining the whole spectrum) has not been done'. It is an unusual thing to do and needs justification. The discussion now begins with outlining the problem with UV fluorescence, but as stated above I think this does not justify ignoring wavelengths between 400-700nm. I understand that the value of this study is that it compared different species using the same technique. However, it is still not fully explained why only long wavelengths were studied and most of the visible spectrum was ignored.

In my original review I said "This study also claims to look at 'sensitivity' at long wavelengths. What the authors did was to determine the response of fish to a set intensity for each wavelength. Much more accurate information could be derived from determining the threshold intensity that evokes a response at each wavelength using a variety of different intensities." The authors acknowledge this by saying "Determining the threshold intensity at each wavelength, as proposed, may provide us more accurate information about fish vision at each wavelength. However, our result showed that spectral sensitivity for each wavelength were different among fish species." While this may be true, it still seems to me that the current data therefore, at best, give an estimate of comparative sensitivity at long wavelengths rather than an accurate measure.

In my original review I made several comments regarding the first paragraph of the introduction. There seemed to be several inaccuracies. These have now been addressed by the authors, but I feel in doing this the authors have produced a very lengthy paragraph, much of which is not directly relevant to the study. Some specific problems also remain;

- In my original review I pointed out that the authors had stated there were only 2 types of photoreceptor, to which I commented "This is only true when considering the outer retina. However, the inner retina contains distinct photoreceptive cells (eg melanopsin)." The authors have changed the text so that the paragraph now says there is also a third class of photoreceptor "photosensitive ganglion cells." This is still inaccurate as other cell types in the retina can also be photosensitive. For example, the first ever non-rod non-cone retinal photoreceptor was described in fish horizontal cells. Also, melanopsin is not the only 'novel' photopigment involved.

- The authors now give quite a lot of information about melanopsin in this first paragraph which does not seem relevant or appropriate to the work presented. I realise that this has probably happened following my original comment, which was only made to point out that the original statement made was wrong. However, I did not mean the authors to include a lot of information that is not really relevant to the current study.
 - The first paragraph also refers to “the African cichlid”. This is not very helpful as there are many cichlids in Africa.
 - This first sentence of the first paragraph lists the 3 photoreceptor types, while the third sentence says “By comparing the activities of different photoreceptor types, wavelength information has been extracted...”. This is misleading as it sounds as if to get wavelength information you compare the output of cones, rods and ganglion cells. What should be said is that colour information is (mainly) gained by comparing the output of cones.
 - It is stated that “cones contain one type of visual pigment”. As it stands this statement is very misleading as most retinas contain several distinct spectral cone types and, as the authors later point out, some cones co-express opsins. The statement needs to be changed to ‘individual cones usually contain a single type of visual pigment’. It also has to be made clear that most species have more than one spectral cone type.
 - Why list the human visual pigments in the introduction to a study on fish?
- Thus, in summary, I do not think the introductory paragraph is very helpful. I assume it should give the basic biology needed to understand the study. However, I think much of it is irrelevant and parts of it are misleading.

I understand that in Figure 1B the bars of various colours underneath the x axis indicate the longest wavelength that evoked a visual response using different measures. However, I do not know what the arrows coming off these bars towards the left indicate. Is the length of the arrow indicative of anything?

In the last paragraph of the Introduction previous work on the animals used in this study is discussed. One sentence starts “Guppy responded to light of 600nm” I think it would be helpful to start this sentence with the phrase ‘In previous studies, guppy responded’.

We are told in the methods that fish were kept under ‘ordinary’ fluorescent light. This is not enough information. One must also be informed of the intensity (in radiometric units) and the spectrum (or at least details of the lights) as both of these can effect visual pigment expression.

In the methods we are told “The time zones during which the experiment was done are as follows: in the first-round assays, 9-11, 750, 900nm; 12-14, 800, 950 nm etc.....”. It is not immediately obvious what this means. I assume what is meant is that the wavelengths 750 and 900nm were tested between 09.00-11.00 etc. If so, could this not simply be stated in this simple manner. Also, this information is only of interest if we know when the light in the maintenance tanks were on and off; we are simply told it was a 14/10hr light dark cycle. This information is important as circadian rhythms could affect spectral sensitivity. Furthermore, although this information is important, I am not sure it is worth describing in the text exactly when what was tested. Perhaps this would be better presented as a table in the supplementary material?

Virtually no details are given of the optomotor experimental setup. Although unusual, it is a common enough technique for a detailed description to be unnecessary. However, I think some details are required. For example, what angle did the stripes subtend at the eye/retina? Or how wide were the stripes and how far from the eye. Without this information how, for example, can we be sure that the lack of an OMR in some species was not due an inability to discriminate the stripes due to lowered acuity?

Table S1 purports to show the longest wavelength which evoked a response in various species of fish in previous studies. In my previous review I questioned how comprehensive the list of studies was. Obviously, I have not checked all papers for all species studies here, but I think that at least some of the values in table S1 are erroneous. Specifically, in my first review I said “Salmonid spectral sensitivity, for example, has been widely studied and some rainbow trout studies are not listed.” As a specific example, the following study (Douglas, R.H., 1983. Spectral sensitivity of rainbow trout (*Salmo gairdneri*). *Revue canadienne de biologie expérimentale*, 42(2), pp.117-122.) showed a response at longer wavelengths than the 640nm stated in the table for rainbow trout. Similarly, the longest wavelength said to invoke a response in goldfish is 660nm. However, the following paper (Schaerer, S. and Neumeyer, C., 1996. Motion detection in goldfish investigated with the optomotor response is “color blind”. *Vision research*, 36(24), pp.4025-4034.) sees a response at 699nm. I fear I may be misunderstanding what sort of studies are included in the table and why.

Regarding Figure 2;

- It is headed “Spectra of monochromatic light from the Okazaki Large Spectrograph (OLS).” What is meant by ‘from the Okazaki Large Spectrograph’?
- The body of the legend say “Spectra were measured separately at $\lambda = 700, 720, 750, 780, 800, 810, 820, 830, 840, 850, 860, 15\ 880, 900, 950,$ and 1000 nm. It is unclear what this means; the spectra are given as solid smooth lines and I assume measurements were taken frequently (e.g. every 1nm). What is written implies they were only taken every 20-30nm.
- The legend goes on to say “The different photon density from the previous paper [23] depended on the cumulative lighting time of the xenon lamp, the light source.” I am afraid I do not know what this means.

The legend of Figure 3 describes the 4 parameters measured in detail. A very similar description is given at the end of section 2.2. Is such repetition necessary? I realise this is most likely a response to my previous comment that the legend was not descriptive enough.

Section 3.1, 2nd para states that rainbow trout and tilapia turned around frequently and refers us to Fig. 3. As far as I can tell, Fig. 3 does not allow us to detect a fish turning.

The following section (3.2) explains well how the data show cavefish do not have an OMR. It might be sensible to merge sections 3.1 & 3.2 and describe all the species that do not have an OMR together?

Figure 4 legend; The title of the legend is “Behavioural response of medaka, goldfish, zebrafish, guppy, three-spined stickleback, and mbuna under the light at 50 nm intervals from 700 nm to 1000 nm”. The first line of the legend itself begins “Behavioural tests were conducted under light of 700, 750, 800, 850, 900, 950, and 1000 nm using 34 goldfish, zebrafish, three-spined stickleback, and mbuna.” These 2 sentences seem to say the same thing. The same comment applies to the legend of figure 5.

Previously I had said “Many of the numerical results given in the text are repetitions of data in tables. This seems unnecessary.” In response to this the authors have removed the tables and reported all the numerical data in the text. I find this makes the text very hard to read and follow. The second paragraph of section 3.3, for example, summarises the overall findings depicted in figure 4 well. The following three paragraph, however, add little to this understanding. I therefore wonder if rather than remove the table, as the authors have done, what should have been done is to keep the tables but remove the details from the text.

Similarly, I think section 3.4 would benefit from less numerical data in the text and should concentrate on a summary of the long wavelength limits of the OMR in the various species.

Throughout the discussion there is confusing and incorrect use of the terms chromophore and opsin. A visual pigment is composed of a light-absorbing chromophore linked to a protein, opsin, which tunes the chromophore. Neither opsin nor the chromophore are a visual pigment. Thus, the Discussion says “The transition from A1 to A2 results in a red shift of spectra which opsins absorb”. The opsin does not absorb light. It is a protein that tunes the light absorbing chromophore. The sentence should probably read something like ‘The transition from A1 to A2 results in a red shift of the visual pigment’s absorption spectrum’. Later it says “... these opsins could absorb light of longer wavelengths. Once more, the opsin does not absorb the light, the visual pigment does. Also “retinal A2 works as the visual pigment...” is incorrect as retinal is not the visual pigment. There are several other examples of similar confusions

Section 4.2 begins with “In past reports, zebrafish and guppy have shown phototaxis under light of wavelength >900 nm, but they did not respond to light >800 nm in the OMR assay.” References need to be provided to substantiate these claims. A few sentences later the authors say “The reason why goldfish could perceive light of longer wavelength is not clear”. It needs to emphasise that this sentence refers to ‘in this study’. Without these insertions it is insufficiently clear when previous studies are being referred to and when statements refer to the present work.

At the end of the Discussion there is mention of the role of melanopsin in the OMR. I just want to repeat what I said at the beginning of this review, melanopsin is not the only non-rod, non-cone visual pigment in fish. There is, for example, VA opsin in horizontal cells.

Decision letter (RSOS-210415.R0)

Dear Dr Matsuo

The Editors assigned to your paper RSOS-210415 "Behavioural red-light sensitivity in fish according to the optomotor response" have now received comments from reviewers and would like you to revise the paper in accordance with the reviewer comments and any comments from the Editors. Please note this decision does not guarantee eventual acceptance.

Please submit your revised manuscript and required files (see below) no later than 21 days from today's (ie 12-Apr-2021) date. Note: the ScholarOne system will 'lock' if submission of the revision is attempted 21 or more days after the deadline. If you do not think you will be able to meet this deadline please contact the editorial office immediately.

on behalf of Kevin Padian (Subject Editor)
openscience@royalsociety.org

Subject Editor Comments to Author:

Comments to the Author:

Thank you for your revisions. As you will see, one reviewer has some substantial remaining concerns. We think these are important to address, and so in your final revision can you please do so. We regret that we would be unable to consider further revision if this one is not sufficient. Best wishes.

Associate Editor Comments to Author:

Comments to the Author:

The reviewers have differing opinions of your work. While you have clearly made some efforts to improve the paper, there remain substantial areas for improvement. We are giving you the benefit of the doubt to give you the chance to revise the paper to address the remaining concerns of the more critical referee. However, as the reviewer notes a number of their criticisms from the first round of review have either not been resolved or have only been partially resolved, we are only going to give you this final opportunity to revise the paper - it is not fair on the reviewers nor on you, as the authors of the paper, for the paper to be in a repeated cycle of reject/resub, reject/resub. If you choose to revise the paper, please make sure you fully address all of the concerns of the referee - if you need an extra week or two to do so, please contact the editorial office for their advice/extension of the revision deadline - because we will not be able to consider the paper further if this reviewer is not satisfied by the changes you make. Please remember that a number of language editing services can be contacted for advice, too, to help ensure the clarity of the scientific story you are endeavouring to tell (<https://royalsociety.org/journals/authors/benefits/language-editing/>). Good luck!

Reviewer comments to Author:

Reviewer: 2

Comments to the Author(s)

In my original review I said "The authors have chosen to only look at long wavelengths (>720nm). Why they have done this is not explained. Most studies of spectral sensitivity include the whole of the 'visible' spectrum. It must be explained early on why this was not done." In their response letter the authors state that this was due to a "technical issue". Specifically, it seems to concern the well-known phenomenon of fluorescence in the UV. This to me still does not explain why wavelengths between the far red and the UV were not examined. There may well be a good reason for only looking at longer wavelengths, but as I said previously 'It must be

explained early on (i.e in the Introduction) why this (examining the whole spectrum) has not been done'. It is an unusual thing to do and needs justification. The discussion now begins with outlining the problem with UV fluorescence, but as stated above I think this does not justify ignoring wavelengths between 400-700nm. I understand that the value of this study is that it compared different species using the same technique. However, it is still not fully explained why only long wavelengths were studied and most of the visible spectrum was ignored.

In my original review I said "This study also claims to look at 'sensitivity' at long wavelengths.

What the authors did was to determine the response of fish to a set intensity for each wavelength. Much more accurate information could be derived from determining the threshold intensity that evokes a response at each wavelength using a variety of different intensities." The authors acknowledge this by saying "Determining the threshold intensity at each wavelength, as proposed, may provide us more accurate information about fish vision at each wavelength.

However, our

result showed that spectral sensitivity for each wavelength were different among fish species."

While this may be true, it still seems to me that the current data therefore, at best, give an estimate of comparative sensitivity at long wavelengths rather than an accurate measure.

In my original review I made several comments regarding the first paragraph of the introduction.

There seemed to be several inaccuracies. These have now been addressed by the authors, but I feel in doing this the authors have produced a very lengthy paragraph, much of which is not directly relevant to the study. Some specific problems also remain;

- In my original review I pointed out that the authors had stated there were only 2 types of photoreceptor, to which I commented "This is only true when considering the outer retina.

However, the inner retina contains distinct photoreceptive cells (eg melanopsin)." The authors have changed the text so that the paragraph now says there is also a third class of photoreceptor "photosensitive ganglion cells." This is still inaccurate as other cell types in the retina can also be photosensitive. For example, the first ever non-rod non-cone retinal photoreceptor was described in fish horizontal cells. Also, melanopsin is not the only 'novel' photopigment involved.

- The authors now give quite a lot of information about melanopsin in this first paragraph which does not seem relevant or appropriate to the work presented. I realise that this has probably happened following my original comment, which was only made to point out that the original statement made was wrong. However, I did not mean the authors to include a lot of information that is not really relevant to the current study.

- The first paragraph also refers to "the African cichlid". This is not very helpful as there are many cichlids in Africa.

- This first sentence of the first paragraph lists the 3 photoreceptor types, while the third sentence says "By comparing the activities of different photoreceptor types, wavelength information has been extracted...". This is misleading as it sounds as if to get wavelength information you compare the output of cones, rods and ganglion cells. What should be said is that colour information is (mainly) gained by comparing the output of cones.

- It is stated that "cones contain one type of visual pigment". As it stands this statement is very misleading as most retinas contain several distinct spectral cone types and, as the authors later point out, some cones co-express opsins. The statement needs to be changed to 'individual cones usually contain a single type of visual pigment'. It also has to be made clear that most species have more than one spectral cone type.

- Why list the human visual pigments in the introduction to a study on fish?

Thus, in summary, I do not think the introductory paragraph is very helpful. I assume it should give the basic biology needed to understand the study. However, I think much of it is irrelevant and parts of it are misleading.

I understand that in Figure 1B the bars of various colours underneath the x axis indicate the longest wavelength that evoked a visual response using different measures. However, I do not

know what the arrows coming off these bars towards the left indicate. Is the length of the arrow indicative of anything?

In the last paragraph of the Introduction previous work on the animals used in this study is discussed. One sentence starts “Guppy responded to light of 600nm” I think it would be helpful to start this sentence with the phrase ‘In previous studies, guppy responded’.

We are told in the methods that fish were kept under ‘ordinary’ fluorescent light. This is not enough information. One must also be informed of the intensity (in radiometric units) and the spectrum (or at least details of the lights) as both of these can effect visual pigment expression.

In the methods we are told “The time zones during which the experiment was done are as follows: in the first-round assays, 9-11, 750, 900nm; 12-14, 800, 950 nm etc.....”. It is not immediately obvious what this means. I assume what is meant is that the wavelengths 750 and 900nm were tested between 09.00-11.00 etc. If so, could this not simply be stated in this simple manner. Also, this information is only of interest if we know when the light in the maintenance tanks were on and off; we are simply told it was a 14/10hr light dark cycle. This information is important as circadian rhythms could affect spectral sensitivity. Furthermore, although this information is important, I am not sure it is worth describing in the text exactly when what was tested. Perhaps this would be better presented as a table in the supplementary material?

Virtually no details are given of the optomotor experimental setup. Although unusual, it is a common enough technique for a detailed description to be unnecessary. However, I think some details are required. For example, what angle did the stripes subtend at the eye/retina? Or how wide were the stripes and how far from the eye. Without this information how, for example, can we be sure that the lack of an OMR in some species was not due an inability to discriminate the stripes due to lowered acuity?

Table S1 purports to show the longest wavelength which evoked a response in various species of fish in previous studies. In my previous review I questioned how comprehensive the list of studies was. Obviously, I have not checked all papers for all species studies here, but I think that at least some of the values in table S1 are erroneous. Specifically, in my first review I said “Salmonid spectral sensitivity, for example, has been widely studied and some rainbow trout studies are not listed.” As a specific example, the following study (Douglas, R.H., 1983. Spectral sensitivity of rainbow trout (*Salmo gairdneri*). *Revue canadienne de biologie expérimentale*, 42(2), pp.117-122.) showed a response at longer wavelengths than the 640nm stated in the table for rainbow trout. Similarly, the longest wavelength said to invoke a response in goldfish is 660nm. However, the following paper (Schaerer, S. and Neumeyer, C., 1996. Motion detection in goldfish investigated with the optomotor response is “color blind”. *Vision research*, 36(24), pp.4025-4034.) sees a response at 699nm. I fear I may be misunderstanding what sort of studies are included in the table and why.

Regarding Figure 2;

- It is headed “Spectra of monochromatic light from the Okazaki Large Spectrograph (OLS).” What is meant by ‘from the Okazaki Large Spectrograph’?
- The body of the legend say “Spectra were measured separately at $\lambda = 700, 720, 750, 780, 800, 810, 820, 830, 840, 850, 860, 15\ 880, 900, 950, \text{ and } 1000 \text{ nm}$. It is unclear what this means; the spectra are given as solid smooth lines and I assume measurements were taken frequently (e.g. every 1nm). What is written implies they were only taken every 20-30nm.
- The legend goes on to say “The different photon density from the previous paper [23] depended on the cumulative lighting time of the xenon lamp, the light source.” I am afraid I do not know what this means.

The legend of Figure 3 describes the 4 parameters measured in detail. A very similar description is given at the end of section 2.2. Is such repetition necessary? I realise this is most likely a response to my previous comment that the legend was not descriptive enough.

Section 3.1, 2nd para states that rainbow trout and tilapia turned around frequently and refers us to Fig. 3. As far as I can tell, Fig. 3 does not allow us to detect a fish turning.

The following section (3.2) explains well how the data show cavefish do not have an OMR. It might be sensible to merge sections 3.1 & 3.2 and describe all the species that do not have an OMR together?

Figure 4 legend; The title of the legend is "Behavioural response of medaka, goldfish, zebrafish, guppy, three-spined stickleback, and mbuna under the light at 50 nm intervals from 700 nm to 1000 nm". The first line of the legend itself begins "Behavioural tests were conducted under light of 700, 750, 800, 850, 900, 950, and 1000 nm using 34 goldfish, zebrafish, three-spined stickleback, and mbuna." These 2 sentences seem to say the same thing. The same comment applies to the legend of figure 5.

Previously I had said "Many of the numerical results given in the text are repetitions of data in tables. This seems unnecessary." In response to this the authors have removed the tables and reported all the numerical data in the text. I find this makes the text very hard to read and follow. The second paragraph of section 3.3, for example, summarises the overall findings depicted in figure 4 well. The following three paragraphs, however, add little to this understanding. I therefore wonder if rather than remove the table, as the authors have done, what should have been done is to keep the tables but remove the details from the text.

Similarly, I think section 3.4 would benefit from less numerical data in the text and should concentrate on a summary of the long wavelength limits of the OMR in the various species.

Throughout the discussion there is confusing and incorrect use of the terms chromophore and opsin. A visual pigment is composed of a light-absorbing chromophore linked to a protein, opsin, which tunes the chromophore. Neither opsin nor the chromophore are a visual pigment. Thus, the Discussion says "The transition from A1 to A2 results in a red shift of spectra which opsins absorb". The opsin does not absorb light. It is a protein that tunes the light absorbing chromophore. The sentence should probably read something like 'The transition from A1 to A2 results in a red shift of the visual pigment's absorption spectrum'. Later it says "... these opsins could absorb light of longer wavelengths. Once more, the opsin does not absorb the light, the visual pigment does. Also "retinal A2 works as the visual pigment..." is incorrect as retinal is not the visual pigment. There are several other examples of similar confusions

Section 4.2 begins with "In past reports, zebrafish and guppy have shown phototaxis under light of wavelength >900 nm, but they did not respond to light >800 nm in the OMR assay." References need to be provided to substantiate these claims. A few sentences later the authors say "The reason why goldfish could perceive light of longer wavelength is not clear". It needs to emphasise that this sentence refers to 'in this study'. Without these insertions it is insufficiently clear when previous studies are being referred to and when statements refer to the present work.

At the end of the Discussion there is mention of the role of melanopsin in the OMR. I just want to repeat what I said at the beginning of this review, melanopsin is not the only non-rod, non-cone visual pigment in fish. There is, for example, VA opsin in horizontal cells.

Reviewer: 1

Comments to the Author(s)

The revised manuscript by Matsuo et al. entitled "Behavioural red-light sensitivity in fish according to the optomotor response" is a valuable addition to the field and in its current state is suitable for publication. The authors conscientiously addressed my critical and rather lengthy comments from the first round of review and I appreciate their efforts. I particularly appreciate the lengths the authors went to clarifying the methods and making them more reproducible, the added sections on ontogeny and chromophores added interesting context, and the figures are much more informative. Figure one in particular is a marked improvement and conveys a lot of really useful details -- my one comment would be to include the data reported here in Fig 1, not just previously reported behavioural and ERG data from the literature (e.g., goldfish are shown to have OMR from wavelengths approximately <650nm but the authors report much longer wavelengths eliciting an OMR response. Overall, I think these data are reported well in the manuscript and publication in the Royal Society Open Science is justified.

===PREPARING YOUR MANUSCRIPT===

===PREPARING YOUR REVISION IN SCHOLARONE===

Author's Response to Decision Letter for (RSOS-210415.R0)

See Appendix B.

RSOS-210415.R1 (Revision)

Review form: Reviewer 2

Is the manuscript scientifically sound in its present form?

Yes

Are the interpretations and conclusions justified by the results?

Yes

Is the language acceptable?

Yes

Do you have any ethical concerns with this paper?

No

Have you any concerns about statistical analyses in this paper?

No

Recommendation?

Accept with minor revision (please list in comments)

Comments to the Author(s)

Many thanks for your careful consideration of all my comments on the previous two versions of the manuscript.

In my opinion, the Introduction is now much improved as unnecessary information has been removed and it has been, briefly, clarified why this study concentrates on the longwave end of the spectrum.

Figure 1 legend this refers to upper and lower columns. Columns are vertical structures. What is meant is 'rows'. The last sentence of the legend says "Previous works on opsin genes and fish vision are summarised in Supplemental Table S1". When I first read this I was confused as to what this table might contain as 'fish vision' is rather a big subject. I did not immediately understand that only the contents of this figure was being referred to. I therefore suggest this sentence be replaced by something like "The original references on which this figure is based are given in Supplemental Table S1"

Figure 2 Panels (B)-(C) are headed "Wavelengths conducted". I do not think 'conducted' is the correct word. Perhaps 'used' would be more appropriate.

In the Methods the length of the fish is given as 3-10cm. When specifying fish size one usually uses 'standard length' (snout to caudal peduncle) but 'total length' (which includes the caudal fin) is also sometimes used. It should be specified whether SL or TL is being used here.

Thank you for including information about the time-of-day various experiments were done in Table S2, as this is potentially important information. However, I have one more minor suggestion. In the text you refer to “time zones”. In English this has a very specific meaning and refers to the time of day at various points on the planet. Of course, this is not what you mean. It would be much clearer for the reader if you were to simply refer to the time-of-day experiments were done and avoid the phrase ‘time zone’.

I appreciate you have taken a detailed description of the parameters out of the legend of Figure 3. In truth I don’t think it is necessary to refer to the Methods for details of the stimuli as this is self-evident and not normal practice. I also do not think you need to say which wavelengths were tested in the legend either as it is obvious from the figure and is also outlined in the Methods. Similar comments apply to the legends of Figs. 4 & 5.

Tilapia data are shown in Figure 3, and this is referred to in section 5.2 of the Discussion. Would it be worthwhile to also refer to the tilapia data in Fig. 3 in the first section of the results (4.1)?

I still find sections such as 4.3 difficult to read due to the plethora of numerical data and statistics in the text. The data in Fig 4 are clear and one can see that in all species ‘duration’ increases with increasing wavelength while the other 3 parameters fall, indicating OMR is lost at longer wavelengths. For me, a simple statement along these lines would suffice, rather than the complex text presented in this section. This is especially true for this section, which is only exploratory for the subsequent experiments. Essentially similar comments apply to the following section (4.4). However, I accept the authors have a problem if presenting the statistics in this way was requested by the other reviewer and will accept what they have done.

In the Discussion it says “As the light source used in this study produced a sequential wavelength of light, a bigger apparatus and a larger aquarium covered a wider wavelength range.” I do not understand what this means. Why does a bigger aquarium result in a larger wavelength range?

The Discussion says “In goldfish, LWS-expressing cones process stimuli to elicit an OMR under light of wavelengths between 409 nm to 699 nm while in adult zebrafish, stimulus occurs under light of 416 nm through to 699 nm [27,28].” It is unclear what this means. The term ‘stimulus occurs’ is especially confusing. Is the following what is meant ‘In goldfish, LWS-expressing cones elicit an OMR at wavelengths between 409-600nm while in adult zebrafish an OMR occurs between 416-699nm’.

The Discussion says “The longest wavelength that provoked an OMR in medaka, goldfish, zebrafish, guppy and stickleback was 800 nm, while being under 780 nm in mbuna. In medaka, the absorption spectrum of LWS pigment was almost zero at 650 nm [9]; in stickleback, 650 nm [10]; in Nile tilapia, 675 nm [11]; and in guppy, 650 nm [12].’ The authors seem concerned that their fish still respond at wavelengths where the absorption of the visual pigment appears to zero. This is, however, not really a problem as the absorption spectrum only seems to approach zero at the wavelengths stated as the curves the visual pigment spectra shown in Fig. 1 are drawn on a linear scale. However, photoreceptors respond over a range on many log units and if visual pigments are displayed using log absorbance it is apparent they could easily mediate longer wavelength sensitivity.

In the last paragraph of the Discussion the authors mention melanopsin may mediate long wavelength sensitivity. This is very unlikely as across all vertebrates melanopsin is maximally sensitive at around 480nm.

Decision letter (RSOS-210415.R1)

Dear Dr Matsuo

On behalf of the Editors, we are pleased to inform you that your Manuscript RSOS-210415.R1 "Behavioural red-light sensitivity in fish according to the optomotor response" has been accepted for publication in Royal Society Open Science subject to minor revision in accordance with the referees' reports. Please find the referees' comments along with any feedback from the Editors below my signature.

Please submit your revised manuscript and required files (see below) no later than 7 days from today's (ie 23-Jun-2021) date. Note: the ScholarOne system will 'lock' if submission of the revision is attempted 7 or more days after the deadline. If you do not think you will be able to meet this deadline please contact the editorial office immediately.

on behalf of Prof Kevin Padian (Subject Editor)
openscience@royalsociety.org

Associate Editor Comments to Author:

The reviewer has identified a number of remaining matters that need to be addressed by you - these appear to be largely textual clarifications and revisions, rather than major conceptual rebuilds; however, you should still pay close attention to the recommendations and requests in preparing your revision. We'll look forward to receiving this in due course.

Reviewer comments to Author:

Reviewer: 2

Comments to the Author(s)

Many thanks for your careful consideration of all my comments on the previous two versions of the manuscript.

In my opinion, the Introduction is now much improved as unnecessary information has been removed and it has been, briefly, clarified why this study concentrates on the longwave end of the spectrum.

Figure 1 legend this refers to upper and lower columns. Columns are vertical structures. What is meant is 'rows'. The last sentence of the legend says "Previous works on opsin genes and fish vision are summarised in Supplemental Table S1". When I first read this I was confused as to what this table might contain as 'fish vision' is rather a big subject. I did not immediately understand that only the contents of this figure was being referred to. I therefore suggest this sentence be replaced by something like "The original references on which this figure is based are given in Supplemental Table S1"

Figure 2 Panels (B)-(C) are headed "Wavelengths conducted". I do not think 'conducted' is the correct word. Perhaps 'used' would be more appropriate.

In the Methods the length of the fish is given as 3-10cm. When specifying fish size one usually uses 'standard length' (snout to caudal peduncle) but 'total length' (which includes the caudal fin) is also sometimes used. It should be specified whether SL or TL is being used here.

Thank you for including information about the time-of-day various experiments were done in Table S2, as this is potentially important information. However, I have one more minor suggestion. In the text you refer to "time zones". In English this has a very specific meaning and refers to the time of day at various points on the planet. Of course, this is not what you mean. It would be much clearer for the reader if you were to simply refer to the time-of-day experiments were done and avoid the phrase 'time zone'.

I appreciate you have taken a detailed description of the parameters out of the legend of Figure 3. In truth I don't think it is necessary to refer to the Methods for details of the stimuli as this is self-evident and not normal practice. I also do not think you need to say which wavelengths were tested in the legend either as it is obvious from the figure and is also outlined in the Methods. Similar comments apply to the legends of Figs. 4 & 5.

Tilapia data are shown in Figure 3, and this is referred to in section 5.2 of the Discussion. Would it be worthwhile to also refer to the tilapia data in Fig. 3 in the first section of the results (4.1)?

I still find sections such as 4.3 difficult to read due to the plethora of numerical data and statistics in the text. The data in Fig 4 are clear and one can see that in all species 'duration' increases with increasing wavelength while the other 3 parameters fall, indicating OMR is lost at longer wavelengths. For me, a simple statement along these lines would suffice, rather than the complex text presented in this section. This is especially true for this section, which is only exploratory for the subsequent experiments. Essentially similar comments apply to the following section (4.4). However, I accept the authors have a problem if presenting the statistics in this way was requested by the other reviewer and will accept what they have done.

In the Discussion it says "As the light source used in this study produced a sequential wavelength of light, a bigger apparatus and a larger aquarium covered a wider wavelength range." I do not understand what this means. Why does a bigger aquarium result in a larger wavelength range?

The Discussion says "In goldfish, LWS-expressing cones process stimuli to elicit an OMR under light of wavelengths between 409 nm to 699 nm while in adult zebrafish, stimulus occurs under light of 416 nm through to 699 nm [27,28]." It is unclear what this means. The term 'stimulus

occurs' is especially confusing. Is the following what is meant 'In goldfish, LWS-expressing cones elicit an OMR at wavelengths between 409-600nm while in adult zebrafish an OMR occurs between 416-699nm'.

The Discussion says 'The longest wavelength that provoked an OMR in medaka, goldfish, zebrafish, guppy and stickleback was 800 nm, while being under 780 nm in mbuna. In medaka, the absorption spectrum of LWS pigment was almost zero at 650 nm [9]; in stickleback, 650 nm [10]; in Nile tilapia, 675 nm [11]; and in guppy, 650 nm [12].' The authors seem concerned that their fish still respond at wavelengths where the absorption of the visual pigment appears to zero. This is, however, not really a problem as the absorption spectrum only seems to approach zero at the wavelengths stated as the curves the visual pigment spectra shown in Fig. 1 are drawn on a linear scale. However, photoreceptors respond over a range on many log units and if visual pigments are displayed using log absorbance it is apparent they could easily mediate longer wavelength sensitivity.

In the last paragraph of the Discussion the authors mention melanopsin may mediate long wavelength sensitivity. This is very unlikely as across all vertebrates melanopsin is maximally sensitive at around 480nm.

===PREPARING YOUR MANUSCRIPT===

===PREPARING YOUR REVISION IN SCHOLARONE===

To revise your manuscript, log into <https://mc.manuscriptcentral.com/rsos> and enter your Author Centre - this may be accessed by clicking on "Author" in the dark toolbar at the top of the

page (just below the journal name). You will find your manuscript listed under "Manuscripts with Decisions". Under "Actions", click on "Create a Revision".

<https://royalsociety.org/journals/authors/author-guidelines/#supplementary-material> to include a suitable title and informative caption. An example of appropriate titling and captioning may be found at https://figshare.com/articles/Table_S2_from_Is_there_a_trade-off_between_peak_performance_and_performance_breadth_across_temperatures_for_aerobic_sc_ope_in_teleost_fishes_/3843624.

Author's Response to Decision Letter for (RSOS-210415.R1)

See Appendix C.

Decision letter (RSOS-210415.R2)

Dear Dr Matsuo,

I am pleased to inform you that your manuscript entitled "Behavioural red-light sensitivity in fish according to the optomotor response" is now accepted for publication in Royal Society Open Science.

on behalf of Prof Kevin Padian (Subject Editor)
openscience@royalsociety.org

Appendix A

Re: Manuscript ID RSOS-201005

Response to Associate Editor and Editor:

Thank you for giving us an opportunity for resubmission. This time we gave it our all and tried to improve the manuscript. The paper was checked by a professional English editing service.

The following is a point-by-point response to the questions and comments.

Response to Reviewer 1:

Thank you for your review of our paper. We have answered each of your points below. This time we tried to improve the manuscript and the paper was checked by an English editing service. We have incorporated changes that reflect the detailed suggestions you have graciously provided.

Technical concerns:

1 P4L32-50: Is the origin of each species known? Are they wild-caught or hatchery/lab reared? Also, stickleback referred to Broughton Archipelago, is this appropriate? We can't know with the information provided.

Response: Medaka was reared in our laboratory. All the other fish species were purchased from the local fish suppliers. According to the fish suppliers, these fish species were hatchery reared. Stickleback originally lived in a coastal area, clear water and had silverish body colour. According to the paper of Flamarique *et al.* (JEB, 2013), sticklebacks living in clearwater (Broughton Archipelago) possessed opsin with A1, whereas fish in red-light shifted lakes (Mayer Lake and Drizzle Lake) had opsin with A2. So that, we referred the data of sticklebacks from Broughton Archipelago. However, as you pointed, sentences in Figure legend were confusing, we changed the corresponding part in legend of Figure 2 (new Figure 1B) as follows:

(P13L11 in revised manuscript clean version) All the chromophores of opsin are A1, but those of goldfish are A2.

2 P4L32: What is the ambient light source used? Strong evidence of lights impacting opsin expression (e.g., Fuller & Claricoates 2011 Molecular Ecology 20, 3321-3335) may influence the assay.

Response: Fish were kept under ordinary fluorescent light. We read the paper of Fuller & Claricoates you suggested. The water was clear and supposed not to affect the expression of opsin. We added the following sentence:

(P3L30) Fish were maintained under ordinary fluorescent light.

3 P4L58: How long were fish light adapted for prior to the behavioural assay? Did you control for circadian rhythms? There is evidence that both opsin expression and activity vary with time of day so this may be of concern for your OMR assay.

Response: Fish were light adapted more than two hours until the behavioural assay. Fish were kept under a fluorescent light except for OMR test. Before and after the OMR test, fish were light adapted.

We did not control for circadian rhythms. However, since behavioural test under one wavelength was conducted in the same time zone for all species, it was certain that there was a difference in wavelength sensitivity at least in that time zone. The LWS expression level and resultant behavioural sensitivity may fluctuate depending on the time of the experiment. But the tendency that the OMR became weaker as the wavelength became longer was consistent throughout this study. So, it is unlikely that OMR will change dramatically due to differences in LWS expression levels.

According to your indication, we clarified the acclimation time, and the time zone when the behavioural test was performed in METHODS. And in DISCUSSION, we stated that it was possible that the different expression level due to the circadian rhythm, not the absorption of LWS or the copy number, somewhat influenced the result.

(P4L9) Fish were light adapted for more than 2 h until the experiments.

(P4L26) The time zones during which the experiment was done are as follows: in the first-round assays, 9-11, 750, 900nm; 12-14, 800, 950 nm; 14-16, 700, 850, 1000nm; in the second round, 9-11, 860 nm; 12-14, 800, 820, 880 nm; 14-16, 780, 840 nm. Behavioural response of cavefish and tilapia was checked at 9-11 under light of 720, 750, 860 nm; 12-14, 800, 810, 820 nm; 14-16, 830, 840, 850 nm.

(P11L13) On the other hand, in fish, opsin expression and activity changed during the day. It is possible that differences in expression levels due to circadian rhythms, rather than LWS copy number or absorption, have had some effect on the results.

4 P4L59-60: What are the spectra for the different wavelengths from the monochromator? I notice many of the wavelengths are similar to those described in previous work (i.e., Matsou et al. 2018) but the photon densities differ. Please provide spectra for all wavelengths used in the assay. A multi-axis figure including spectra of ambient light and monochromator light would be useful.

Response: As suggested, spectra for all wavelengths from the monochromator used in this work and ambient light are shown in new Figure 2. The different photon density from the previous paper (Matsuo *et al.*, 2018) depended on the cumulative lighting time of the xenon lamp.

5 P5L5: Fish were acclimated to the assay for 30 seconds, although it is not clear if they were transferred to a new behavioral arena or if the OMR rig was lowered into the aquaria. If they were physically removed, e.g., by hand net, then 30 seconds does not seem adequate for the fish to recover equilibria and resume normal behaviours. Please elaborate.

Response: Whether acclimation time of 30 seconds was best for fish to resume normal behaviour is unclear, our method elicited OMR in eight fish species. It was possible that the longer the acclimation time, the better the response of the fish.

Before the experiments under all the wavelength of light, fish were put in separated cups. Until the end of all the behavioural tests, we kept fish isolated.

Detailed procedure of OMR test are as follows:

1. Fish was gently transferred with water from the cup to the testing apparatus.
2. We checked the OMR apparatus, opened the shutter of the monochromatic light source, started video recording, and then turned ceiling lights off.
3. We counted 30 seconds for acclimation without rotating the stripes of OMR apparatus.
4. We started rotating the stripes clockwise and kept moving for 30 seconds.
5. We switched the rotating direction. The stripes moved counterclockwise for 30 seconds.
6. The stripes moved clockwise for 30 seconds.
7. The stripes moved counterclockwise for 30 seconds.
8. We stopped rotating the stripes, turned on the ceiling light, closed the shutter of the monochromatic light source, and stopped video recording.

According to your suggestion, this section was rephrased as follows:

(P4L13) Before all experiments, we put fish in separate cups. Until the end of all the behavioural tests, we kept fish isolated. First, a fish was gently transferred with water from the cup to the testing apparatus. We checked the OMR apparatus, opened the shutter of the monochromatic light source, started video recording, and then turned ceiling lights off. We counted 30 s for acclimatization without rotating the stripes of the OMR apparatus, followed by the subsequent OMR test. During the behavioural test the drum rotated, switching direction every 30 s two or three times. After stopping the rotation, we turned on the ceiling light, closed the shutter of the monochromatic light source, and stopped video recording. After the test, fish were light adapted again.

Following your suggestion, we added three sentences in DISCUSSION as below:

(P9L15) And whether 30 s of acclimation was best for fish to resume normal behaviour was unclear. It is possible that the longer the acclimation time, the better the fishes' response. However, our method elicited OMR in eight fish species.

6 P6L9-14: Can you elaborate on why you excluded tetra and corydoras from later analyses? Here you describe them as OMR positive and refer to video files but there is no follow-up.

Response: We focused on fish species widely used as model animals.

This section was rephrased as follows:

(P5L26) First, we performed an OMR assay at 700 nm or 720 nm. Eight fish species (goldfish, zebrafish, guppy, medaka, stickleback, mbuna, tetra, and corydoras) followed the stripes. Based on our manual observations, all eight fish species were OMR positive (examples of OMR assay of tetra and corydoras are in Video files S1 and S2). Among them, six fish species (goldfish, zebrafish, guppy, medaka, stickleback, and mbuna), widely used as model animals, were subjected to further behavioural assay.

7 P6L31-41: Please provide statistics/p-values supporting the statements herein.

Response: As indicated, we added statistics/p-values as follows:

(P6L4) Mexican cavefish have degenerated eyes. We tested their behaviour under light of 720, 750, 800, 820, and 860 nm. Values of the four parameters were computed and are summarized in Figure 3. Under all wavelengths, the Delay parameter fluctuated and had no significant difference (one-way repeated measures ANOVA, $F_{(4, 20)} = 0.6431$, $P = 0.6381$). Duration was between 0.202 (750 nm) and 0.304 (820 nm) ($P > 0.05$ in all comparisons).

From 720 nm to 860 nm, Angular Velocity was maintained (one-way repeated measures ANOVA, $F_{(4, 20)} = 0.7193$, $P = 0.5888$). Distance at any wavelength was negative, which meant cavefish never followed the signals (one-way repeated measures ANOVA, $F_{(4, 20)} = 0.5802$, $P = 0.6804$). Based on these four parameters, we judged cavefish to be OMR negative at all wavelengths tested.

8 P7L25: p-values are used but please report the F-statistic or any other appropriate statistic for the parametric/non-parametric tests used.

Response: As suggested, statistics (F or t) have been added as follows:

(P6L32) The Duration for these three fish species attenuated to longer wavelength (stickleback, one-way repeated measures ANOVA, $F_{(2.45, 12.26)} = 8.1019$, $P = 0.0042$; medaka, 800 > 950: $t = 2.8823$, $P = 0.0322$; guppy, $P > 0.05$ in all comparisons). In mbuna, Duration at 700 nm and 750 nm was 0.52 and 0.43, respectively, dropped at 800 nm, and showed little change up to 1000 nm ($P > 0.05$ in all comparisons).

9 P8L31: What about stickleback 800>820 comparison?

Response: In comparison between 800vs820 of Distance in sticklebacks, $t=2.2317/p=0.5320$. We rewrote as follows:

(P8L1) Distance at 800 nm did not differ from the value at 820 nm ($t=2.2317/p=0.5320$). Stickleback lost their vision around 820 nm.

10 P8L49-51: Are you comparing data from the first round and second round? Please clarify.

Response: We did not compare data of two rounds since we adopted repeated measures design. This section was rephrased as follows:

(P8L14) All four parameters did not change under light from 780 nm through to 880 nm. In second-round OMR, mbuna perceived neither the upper nor the lower wavelength tested. When we consider the first- and second-round results together, the threshold wavelength for this species must be between 750 nm and 780 nm.

11 P9L59-P10L8: The speed of the OMR apparatus is clearly important. Does this need to be empirically calibrated per species, or perhaps per individual based on size?

Response: The best speed of stimuli, the bandwidth of stripes, size of apparatus to induce OMR might vary by fish species. However, in this paper, our goal was to evaluate fish behavioural response by a unified procedure in a comparable way. Therefore, we did not change experimental conditions through all the behavioural tests. We added the following sentences:

(P4L21) In all the behavioural tests, stripes rotated at 10 rpm (60 degrees/second).

(P9L14) The best speed of stimuli, the best bandwidth of stripe, size of apparatus to induce OMR might vary by fish species. However, our method elicited OMR in eight fish species. To compare spectral sensitivity among fish species, we did not change the experimental conditions for each species and evaluated their behavioural response using a single consistent procedure.

12 P10L34: reference to “not fully-grown fish”, read: juvenile. Ontogeny is known to affect vision. Some description of the limitations of the assay for larger fish and the stage of development of fishes used in the present study is warranted. Same goes for retinal and dehydroretinal and its impact on visual pigments. Some species use both vitamins A1 and A2 depending on season, life history. The manuscript briefly described A1/A2 but a more detailed description of this phenomenon is warranted.

Response: We sincerely accepted comments of Reviewer 1. As suggested, we added following sentences:

(P9L31) Development is known to affect vision. For example, salmon changes its A1/A2 chromophore seasonally [35], and the shift of the A1/A2 ratio correlates with the fish’s life cycle [36]. The transition from A1 to A2 results in a red shift of spectra which opsin absorbs [37,38]. Besides a change of chromophore usage, fish express a different subset of opsins at different developmental stages [11,39–41]. In single cones of salmon, the youngest fish (yolk-sac alevins) express opsin SWS1, whereas older juveniles (smolts) express SWS2 predominantly. In this way, opsin and its chromophore usage change, depending on fish ontogeny. Sensitivities to the deep-red light of adult fish possibly differ from current results.

13 Spec of opsins from table 1, figure 2... maxima only tells us so much, what is half-bandwidth, or better yet, provide the full spectra often reported in the studies you refer to. Many assays (e.g., microspectrophotometry) do not include infrared in their recordings but some may? What about non-visual opsins playing a role in sensitivity. Melanopsin contributes to involuntary pupillary reflex in humans.

Response: We sincerely accepted comments of Reviewer 1. In Figure 2 (new Figure 1B), absorption spectra were depicted instead of absorption maxima. In Table 1 and Figure 2 (new Table S1 and Figure 1B), we summarized the longest wavelength to which animal responded. Some assays as OMR, phototaxis and cardiac conditioning experiments included near-infrared light. Besides opsin of rods and cones, melanopsin was included in Table 1, Figure 2 (new Table S1 and Figure 1B), INTRODUCTION and DISCUSSION as follows:

(P2L27) Photosensitive ganglion cells contain the photopigment melanopsin. Melanopsin is involved in the regulation of circadian rhythms and pupil light reflexes [2]. In teleost, such as zebrafish and the African cichlid, melanopsin (OPN4) was identified [3–6]. In humans, melanopsin appears to enhance the perception of brightness [7] and colour [8].

(P11L32) In humans, melanopsin is involved in the perception of brightness and colour [7,8]. The role of teleost melanopsin is unclear, but this gene is a candidate for detecting deep-red and NIR light. There could be other interpretations of the role that rhodopsin plays in the perception of longer wavelength light. In our previous work using light-adapted medaka, we found that LWS, not rhodopsin, activated OMR [22]. However, it was unclear whether the rods did not activate an OMR under NIR light in other fish species after light adaptation. Future studies testing OMR using fish with knock-out LWS, rhodopsin, or melanopsin genes will solve this question.

14 Tables 3 & 4: If displayed in a table like this, please fill out all the appropriate statistics regardless of significance. Or provide these data in a supplementary table. Bold or highlight

those that are significant if you want to emphasize those data. Perhaps bold/highlight the greatest difference/threshold data you describe in text.

Response: As suggested by Reviewer 1 and Reviewer 2, Tables 3 and 4 were omitted.

15 Please add the data from cavefish on the plots from Fig 3 to have a ready comparison on an OMR negative fish.

Response: As suggested, we combined the data of Figs 3 and 4 into new Figure 3.

16 Figure 5 & 6: What do the broken lines indicate? Also, why were guppy and medaka not tested at 900 and 1000 nm? The exclusion of these without explanation is concerning.

Response: In guppy and medaka, behavioural tests at 900 nm were not conducted, and we connected two values of 850 nm and 950 nm with dotted lines. But in new Figures of revised version, we no more use a solid/dotted line to connect values.

The first-round OMR test was done to narrow the range of wavelength of detailed investigation in the second round. In guppy, all four parameters fell at 850nm. Therefore, we skipped 900 nm and 1000 nm in the first-round OMR assay, and focused around 840 nm and 860 nm in the second round of OMR test, due to the schedule of the experiment.

In medaka, previous works showed that medaka was not OMR-positive under light of >830nm (Homma *et al.*, BMC Gent. 2017, Matsuo *et al.*, Biol. Open 2018). We did not test 900 nor 1000 nm in the first round and focused around 820 nm and 840 nm in the second round. We rewrote METHODS as follows:

(P4L21) We conducted two rounds of OMR assay. The first round was conducted to narrow the range of wavelength of detailed investigation to be conducted in the second-round test. In the first-round behavioural assays, fish were tested under the light at 50 nm intervals from 700 nm to 1000 nm. In the second round, we checked fish behaviour under the light at 20nm intervals from 780 nm to 880 nm..... In guppy, all four parameters fell at 850nm. Therefore, we skipped 900 nm and 1000 nm in the first-round OMR assay, and focused around 840 nm and 860 nm in the second-round OMR test (780, 820, 840, 860 and 880 nm), due to the schedule of the experiment. In medaka, previous works showed that medaka was not OMR positive under light of >830 nm [22,23]. We did not test 900 or 1000 nm in the first round and focused on 820 nm and 840 nm in the second round (780, 800, 820, 840 and 880 nm). All the behaviour during the OMR test was.

17 Figure 7: Data are depicted for a medaka LWS knock-out, where were these data derived from? The red box around medaka LWS, how did you arrive at this box, was it empirically tested? The lack of transparency here is concerning.

The statement that the maximum wavelength eliciting OMR correlates with LWS spectra is tenuous at best. That claim is made in the abstract as if it were empirically tested but that is not the case.

Response: We sincerely accepted comments of Reviewer 1. Data of LWS knock-out medaka was derived from our previous papers, Homma *et al.*. As suggested, we removed both red box and data of LWS knock-out medaka. Also, we rewrote P2L51 in SUMMARY (of previous version of the manuscript) as follows:

(P2L12 in revised manuscript clean version) Fish opsin repertoire enabled the perception of red light. In contrast, the copy number of *long-wave-sensitive (LWS)* genes did not necessarily improve red-light sensitivity.

Copy-edit, clarity, and other minor comments:

18 P2L39: “the visibility” is awkward, alternatives: visual ability, vision, visual sensitivity

Response: As suggested, “the visibility” has been rewritten to “visual sensitivity” (P2L5).

19 P2L44-45: “...sake of no or reckless swimming” consider rewording. I am not sure what you mean by “reckless swimming”

Response: As suggested, this sentence has been rephrased as follows:
(P2L7) some species were not appropriate for the OMR test because they either stayed still or changed swimming direction frequently.

20 P2L51: “absorption spectra” these data are not shown, you only describe max sensitivity.

Response: Based on #20 and also #17, this sentence has been rephrased as follows:
(P2L12) Fish opsin repertoire enabled the perception of red light.

21 P2L53: multiplication should read duplication

Response: As suggested, this sentence has been rephrased properly as follows:
(P2L13) The duplication of *LWS*

22 P2L54: “might aid not increasing” grammar, reword

Response: As suggested, this sentence has been rephrased properly as follows:
(P2L15) might not aid increasing

23 P3L14: The universal paradigm of one rod opsin may be in question, see: Musilova et al. Science. 2019; 364(6440): 588–592.

Response: We appreciate helpful suggestion. This sentence has been rephrased as follows:
(P2L32) Some deep-sea teleost lineages as Myctophidae, Stylephoridae and Diretmidae have multiple rod opsins expressed [9].

24 P3L15-16: “...every cone includes several type of cone visual pigments...” as written this statement incorrectly implies opsin co-expression occurs in every cell.

Response: We appreciate helpful suggestion. This sentence has been rephrased as follows:
(P2L34) Cones contain one type of visual pigment with different absorption maxima (λ_{max}).... In rainbow trout, cichlid and anemonefish, some photoreceptor cells co-express spectrally distinct opsins [11–15].

25 P3L31: "...the spectral sensitivity..." omit "the"

Response: As suggested, "the spectral sensitivity" has been rewritten to "spectral sensitivity" (P3L5).

26 P3L37: "The schooling behavior..." implies you've introduced schooling behavior as a concept prior. Omit "the" or reword the sentence. "This Mexican cavefish" omit "this"

Response: As suggested, "The schooling behavior..." has been rewritten to "Schooling behavior", and "This Mexican cavefish" to "Mexican cavefish" (P3L9).

27 P3L31-42: Consider rewording this entire section for clarity. The last two sentences are particularly difficult to understand.

Response: We appreciate helpful suggestion. This section has been rewritten as follows:
(P3L5) To study spectral sensitivity, we used zebrafish, goldfish, guppy, medaka, cichlid (Nile tilapia and mbuna), three-spined stickleback, bronze corydoras, Mexican cavefish, glowlight tetra, Japanese striped loach, rainbow trout, and Senegal bichir (Figure 1). All the fish species of this work belong to the class Actinopterygii. Mexican cavefish, also known as blind cave tetra, have degenerated eyes and have lost schooling behaviour [19]. We introduced Mexican cavefish as a negative control in this study.

28 P3L45: Consider writing about how the diversity of opsins is driven mostly by tandem duplication.

Response: We appreciate your helpful suggestion. But as suggested by Reviewer 2, this paragraph was removed (P3L10).

29 P3L60: "Besides the OMR, fish vision was examined..." as written this sentence suggests the present study used ERG, phototaxis, OKR. Reword. P4L4: "...in guppy as longest", as longest what? Unclear.

Response: As suggested by Reviewer 1 and Reviewer 2, this section has been rephrased as follows:

(P3L12) Besides OMR, fish vision has been examined in several ways, including electroretinography (ERG), phototaxis, and optokinetic response (OKR). We summarized the longest wavelength to which animals responded (Figure 1, Table S1). Guppy responded to light of 600 nm as measured using the OMR assay.....

30 P4L42: “till” should read “until” this misspelling occurs in multiple instances throughout the manuscript.

Response: As suggested, this word was replaced properly.

31 P4L49: How many animals were used for each species? Why did you use fewer animals for these species? Provide justification.

Response: We used six animals for medaka, goldfish, zebrafish, guppy, three-spined stickleback, mbuna, Mexican cavefish and Nile tilapia. We used one each of Senegal bichir, Japanese striped loach, and rainbow trout. Bichir and loach seldom swam during an assay, and we could not assess their behaviour. Rainbow trout swam violently, changed swimming direction frequently, and we could not assess its behaviour, either. Two fish of glowlight tetra and three of bronze corydoras were used in OMR assay. These two species followed the stripes and were OMR-positive. In further analyses, we focused on fish species widely used as a model animal. This section was rephrased as follows:

(P4L4) Six each of medaka, goldfish, zebrafish, guppy, three-spined stickleback, mbuna, Mexican cavefish, and Nile tilapia were subjected to the OMR test. One each of Senegal bichir, Japanese striped loach, and rainbow trout, two glowlight tetra, and three bronze corydoras were used.

The first paragraph in Results was rewritten as follows:

(P5L26) First, we performed an OMR assay at 700 nm or 720 nm. Eight fish species (goldfish, zebrafish, guppy, medaka, stickleback, mbuna, tetra, and corydoras) followed the stripes. Based on our manual observations, all eight fish species were OMR positive (examples of OMR assay of tetra and corydoras are in Video files S1 and S2). Among them, six fish species (goldfish, zebrafish, guppy, medaka, stickleback, and mbuna), widely used as model animals, were subjected to further behavioural assay.

32 P5L10: “...allocated ID...” is a strange way to phrase this. Is the purpose of this statement to imply the person conducting downstream analyses was “blind” e.g., unaware of the species they were analyzing to reduce bias?

Response: ID was used in statistical analyses of repeated measures way. In “Statistics” of METHODS (P5L18), we clarified that we adopted the repeated measures way and removed this sentence.

33 P5L14-17: “...exclude the noise. Guppy male with...” a little awkward, consider rewording e.g., “...exclude the noise, for example, tracking was made difficult when a male guppy’s long tail fin disturbed the water surface.”

Response: We appreciate the helpful suggestion. This sentence has been rephrased according to your example (P4L37).

34 P5L23-31: The use of parentheses here is unnecessary. Also, the description of distance as “plus” and “minus” is confusing. Suggestion: “4. Distance: the overall swimming distance a

fish swims in the direction of the rotating apparatus. When a fish swims in the opposite direction that distance is subtracted from the overall distance for the trial.”

Response: According to your suggestion, this sentence has been rephrased as follows:

(P5L3) The data were converted into x–y co-ordinates by UMATracker software [23]. With sets of co-ordinates, we calculated four parameters: (1) Delay, the elapsed time (s) until a fish started to follow the pattern after switching the rotating direction of the drum; (2) Duration, the ratio of the time during which a fish followed the striped pattern divided by the total testing time, (3) Angular Velocity, the speed (degrees/s) at which a fish swam to chase the stripes, and (4) Distance, the overall swimming distance (rounds) a fish swam in the direction of the rotating apparatus. When a fish swam in the opposite direction, that distance was subtracted from the overall distance for the trial.

35 P5L30: misplaced “

Response: As suggested, this section was properly rephrased as follows:

(P5L13) “anovakun” version 4.6.2,” -> “anovakun” version 4.6.2

36 P6L5: Header, “applicable” and “inapplicable” is awkward. Use instead included/excluded? Or initial OMR test, and you found a number that were OMR negative.

Response: As suggested, we rephrased this header as follows:

(P5L25) Eight fish species were OMR-positive in first-round test.

37 P6L28: NIR is not defined in-text

Response: In INTRODUCTION, we defined NIR as follows:

(P3L20) Phototaxis has been observed in near-infrared (NIR) light in zebrafish, guppy, and Nile tilapia.

38 P6L48: “These fish were further testes their behavior.” testes should be tested, also the grammar of this sentence is incorrect. The first three paragraphs of 3.3 should be restructured for better grammar/clarity.

Response: We appreciate helpful suggestion. We modified “testes” to “tested” and rewrote these paragraphs as follows:

(P6L14) Goldfish, zebrafish, guppy, medaka, three-spined stickleback, and mbuna followed the rotating pattern at 700 nm. We conducted OMR assays using these six fish species under light of wavelengths between 700 nm and 1000 nm, based on our previous study [23] (first-round OMR test, Figure 4).

We calculated four parameters to judge each fish's response: Delay, Duration, Angular Velocity, and Distance (Figure 4). Overall, Delay increased, and Duration, Angular Velocity, and Distance decreased towards longer wavelengths. Standard errors of the Delay parameter were rather large, fluctuated with wavelength, and seemed to depend on chance. A sharp drop in Duration, Angular Velocity, and Distance indicated a loss of OMR behaviour.

When an animal gradually lost its vision, Delay increased while Duration, Angular Velocity, and Distance decreased.

Delay increased towards longer wavelengths (zebrafish, $750 < 1000$: $t = 11.7866$, $P = 0.0197$; goldfish, $750 < 950$: $t = 42.1836$, $P = 0.0006$; mbuna, $800 < 1000$: $t = 6.4248$, $P = 0.0285$).

39 P6L56-P7L8: “soared” should be “increased”, “went down” should be “decreased”. “Steep drop in the graphs” is a strange way of describing the data, consider rewording. This applies to later examples throughout the text where you refer to the “graphs of...” or “the line graph of...”

Response: We appreciate the helpful suggestion. These sentences have been rephrased as follows:

(P6L18) We calculated four parameters to judge each fish's response: Delay, Duration, Angular Velocity, and Distance (Figure 4). Overall, Delay increased, and Duration, Angular Velocity, and Distance decreased towards longer wavelengths. Standard errors of the Delay parameter were rather large, fluctuated with wavelength, and seemed to depend on chance. A sharp drop in Duration, Angular Velocity, and Distance indicated a loss of OMR behaviour. When an animal gradually lost its vision, Delay increased while Duration, Angular Velocity, and Distance decreased.

40 P7L5: “animal” reword, instead “an animal” or “fish”

Response: As suggested, “animal” was changed to “an animal” (P6L22).

41 P7L15-28: this is the first mention of a “ratio” in text. What is this referring to? Describe in methods first.

Response: As suggested, in “Behavioural Test” of METHODS (P4L33), we described about “ratio” as follows:

(P5L5) Duration, the ratio of the time during which a fish followed the striped pattern divided by the total testing time,

42 P7L31: the speed of the apparatus belongs in the methods.

Response: As suggested, the following sentence was removed.

43 P7L42: introducing a new metric, rounds/test how is this different from distance? Describe in more detail in the methods section.

You ran an initial test for OMR and then a subsequent test for NIR sensitivity using the OMR assay. That could be clarified in the methods section.

Response: “rounds/test” was fixed to “rounds” (P7L8). In METHODS, we added following sentences:

(P4L21) We conducted two rounds of OMR assay. In all the behavioural tests, stripes rotated at 10 rpm (60 degrees/second). The first round was conducted to narrow the range of

wavelength of detailed investigation to be conducted in the second-round test. In the first-round behavioural assays, fish were tested under the light at 50 nm intervals from 700 nm to 1000 nm. In the second round, we checked fish behaviour under the light at 20 nm intervals from 780 nm to 880 nm.

44 P8L26: Report the p-values and F-statistics.

Response: As suggested, this sentence was rephrased as follows:

(P7L32) Statistics of Delay and Duration gave *P*-values under 0.05 (one-way repeated measures ANOVA, Delay: $F_{(5, 25)} = 6.1611$, $P = 0.0008$; Duration: $F_{(5, 25)} = 7.7186$, $P = 0.0002$).

45 P8L42: "(840:880)" format differs from above. Be consistent.

Response: As suggested, "(840:880)" was changed to "(840 > 880)" (P8L8).

46 P8L55-P9L5: This section may be more appropriate in the discussion.

Response: As suggested, this section was moved to 4th paragraph in DISCUSSION (P10L3).

47 P9L3: "almost touched to horizontal axis" reword, describe the data, not the line/graph itself.

Response: As suggested, this sentence was rephrased as follows:

(P10L13) Distance also dropped to around zero.

We moved this sentence to 4th paragraph in DISCUSSION, as in #46 of Reviewer 1.

48 P9L26: "could lost" reword

Response: As suggested, we rephrased as follows:

(P8L29) guppy may have lost vision around these wavelengths.

49 P9L40-41: "Moreover, ..." This sentence is confusing, what are you trying to say here?

Response: This sentence was rephrased as follows:

(P9L6) from the viewpoint of animal ethics, our procedure is an ethical way to examine the vision of animals. We assessed animal behaviour without administering drugs to the fish or restraining their movements.

50 P9L54: What is scrambled and loitered as it relates to fish behaviour? Please reword/clarify.

Response: We appreciate helpful suggestion. This sentence has been rephrased as follows:

(P9L12) fish sometimes stopped following the stimuli.

51 P10L11: Reword this first sentence.

Response: As suggested, this sentence was rephrased as follows:

(P10L1) All four parameters helped to distinguish OMR behaviour from non-OMR behaviour.

52 P10L14: "...affected by chance..." was it or was it not?

Response: As this sentence was confusing, we removed (P10L8).

53 P10L17-18: Was distance the most useful tool? You described all four as useful above. This is confusing.

Response: We appreciate helpful suggestion. We omitted these sentences (P10L8).

54 P10L20-21: "The number over zero..." I do not understand what this sentence is trying to convey.

Response: We removed this sentence. Instead, as suggested in #46, P8L55-P9L5 of the manuscript of the previous version was inserted (P10L3).

55 P10L29: "not fully-grown fish" e.g., juvenile?

Response: As suggested, "not fully-grown fish" was changed to "juvenile" (P9L30).

56 P11L26 and P11L45: "et al" should read "et al." al. is the abbreviated form of alia

Response: As suggested, "et al" was changed to "*et al.*" (P10L31).

57 P11L28: what is meant here by "extend"?

Response: The word "extend" was changed to "hold" (P10L32).

58 The format of figure captions is inconsistent.

Response: As suggested, we rephrased the figure captions as follows:

(P12L29) Figures 1, 2 and 7 (new Figure 1). Fish species in this work.

(A) Phylogenetic relationships of fish. A tree was created based on previous works [62–64].

(B) Visual sensitivity of fish. The absorption spectra of opsin (upper column) and electrophysiological and behavioural response (lower column) of fish are summarized. A horizontal axis shows the wavelength of light. The spectrum of SWS1 is depicted in violet,

SWS2 in blue, RH2 in green, LWS in red, and RH1 and OPN4 in black. The wavelengths of light that have evoked a behavioural or electrophysiological response in fish are depicted as bars and arrows. A bar indicates the longest wavelength that has invoked fish response; phototaxis (green), ERG (orange), OMR (blue), and foraging experiment and cardiac conditioning experiments (black). The wavelengths to which fish responded in this work are shown as magenta bars and arrows. Guppy has one additional LWS with unknown absorption spectrum. Guppy LWS-1 is LWS-1/180Ser. Mbuna has one LWS with unknown absorption spectrum. All the chromophores of opsin are A1, but those of goldfish are A2. Previous works on opsin genes and fish vision are summarized in Supplemental Table S1.

Figures 3 and 4 (new Figure 3). Behavioural response of Nile tilapia and Mexican cavefish under the light from 720 nm to 860 nm.

Behavioural tests were conducted under light of 720, 750, 800, 810, 820, 830, 840, 850, and 860 nm for Nile tilapia (red line), and 720, 750, 800, 820, and 860 nm for Mexican cavefish (black line). In all figures, the horizontal axis indicates the wavelength. The OMR was quantified using the four parameters. (A) Delay. The seconds until a fish turned its body axis and followed the rotating stripes. (B) Duration. The ratio of the time during which a fish followed the stripes divided by the total testing time. A ratio of one means that a fish followed the pattern all the time during the test. (C) Angular Velocity. The speed (degrees/s) at which a fish followed the pattern. (D) Distance. Total Distance (rounds) a fish swam during the test. Values represent means. Error bars are standard errors (SEs).

Figure 5 (new Figure 4). Behavioural response of medaka, goldfish, zebrafish, guppy, three-spined stickleback, and mbuna under the light at 50 nm intervals from 700 nm to 1000 nm in the first-round OMR assay.

Behavioural tests were conducted under light of 700, 750, 800, 850, 900, 950, and 1000 nm using goldfish, zebrafish, three-spined stickleback, and mbuna. Medaka and guppy were tested under light of 700, 750, 800, 850, and 950 nm. In all figures, the horizontal axis indicates the wavelength. The OMR was quantified using the four parameters. (A) Delay. (B) Duration. (C) Angular Velocity. (D) Distance. Definition of the four parameters is as in Figure 3. Values represent means. Error bars are standard errors (SEs). We summarized the graphs according to fish species in Supplemental Figure S1.

Figure 6 (new Figure 5). Behavioural response of medaka, goldfish, zebrafish, guppy, three-spined stickleback, and mbuna under the light at 20 nm intervals from 780 nm to 880 nm in the second-round OMR assay.

Behavioural tests were conducted under light of 780, 800, 820, 840, 860, and 880 nm using goldfish, zebrafish, three-spined stickleback, and mbuna. Guppy were tested under light of 780, 820, 840, 860, and 880 nm. OMR assays using medaka were performed under light of 780, 800, 820, 840, and 880 nm. In all figures, the horizontal axis indicates the wavelength. The OMR was quantified by the four parameters. (A) Delay. (B) Duration. (C) Angular Velocity. (D) Distance. Definition of the four parameters is as in Figure 3. Values represent means. Error bars are standard errors (SEs). We summarized the graphs according to fish species in Supplemental Figure S2.

59 Figures 1 and 2 could be combined for a single visually appealing figure.

60 Figure 2 and 7 appear to convey similar information, perhaps combine them. A multi-panel figure with phylogeny (as mentioned above) would be a great summary figure.

Response to #59 and #60: We appreciate helpful suggestion. We combined Figures 1, 2 and 7 into new Figure 1.

61 Figures 3-6 need better labelling and provide more detailed figure caption for Figs.3 and 4

Response: As suggested in #15, Figures 3 and 4 were combined into new Figure 3. Figure title and caption were rephrased as in #58.

Response to Reviewer 2:

Thank you for your review of our paper. We have answered each of your points below. We have incorporated changes that reflect the detailed suggestions you have graciously provided.

1 The authors have chosen to only look at long wavelengths (>720nm). Why they have done this is not explained. Most studies of spectral sensitivity include the whole of the 'visible' spectrum. It must be explained early on why this was not done.

Response: It's just due to a technical issue. We also attempted to investigate the wavelength sensitivity in the ultraviolet region by this method, but still uncertain about the result. We could not detect the difference in wavelength sensitivity between wild-type and SWS1 KO medaka (unpublished). We could not conclude whether there was no difference or whether fluorescence (long-wavelength light) contamination due to UV irradiation affected the result. Since red light did not have such a problem and stable results could be obtained, we focused on long-wavelength light in this study.

This point was described in DISCUSSION as follows:

(P9L1 in revised manuscript clean version) In this study, we compared red-light sensitivity among fish species. In OMR assays in the ultraviolet region, we needed to avoid fluorescence contamination (our unpublished observation). Since red light did not have such a problem and stable results could be obtained, we assessed visual sensitivity for long-wavelength light.

2 This study also claims to look at 'sensitivity' at long wavelengths. What the authors did was to determine the response of fish to a set intensity for each wavelength. Much more accurate information could be derived from determining the threshold intensity that evokes a response at each wavelength using a variety of different intensities.

Response: Determining the threshold intensity at each wavelength, as proposed, may provide us more accurate information about fish vision at each wavelength. However, our result showed that spectral sensitivity for each wavelength were different among fish species. It was clear that fish species which were OMR-positive had a low threshold and species which did not respond the stimuli had a high threshold.

3 The first paragraph of the introduction sets out basic visual concepts. Unfortunately, it is possible to take issue with many statements.

- “There are two types of photoreceptor.....” This is only true when considering the outer retina. However, the inner retina contains distinct photoreceptive cells (eg melanopsin)

Response: We appreciate helpful suggestion. This section was changed as follows: (P2L22) In the retinae of vertebrates, there are three classes of light-sensitive cells: rod photoreceptors, cone photoreceptors, and photosensitive ganglion cells.. Rod photoreceptor cells are responsible for scotopic vision, the vision working under dim light. Photopic vision, the vision working under daylight, is mediated by cone photoreceptors. Photosensitive ganglion cells contain the photopigment melanopsin. Melanopsin is involved in the regulation of circadian rhythms and pupil light reflexes.

- “Rods contain a single visual pigment” This is not true for amphibia or many deep-sea fish.

Response: As suggested, this section was rephrased as follows: (P2L31) Rods of most vertebrates contain a single rod visual pigment (rhodopsin; RH1). Some deep-sea teleost lineages such as Myctophidae, Stylephoridae, and Diretmidae have multiple rod opsins expressed [9]. Amphibia have two types of rods: typical rods and green rods whose absorbance peaks in the blue part of the spectrum [10].

- “Integration of (cone) signals enable animals to discriminate colour....” Not if the ‘integration is simply additive.

I suggest this paragraph be adjusted to be less generalised.

Response: As suggested, this section was rephrased as follows: (P2L23) Photon signals activate one or more types of photoreceptor cells. By comparing the activities of different photoreceptor types, wavelength information has been extracted in colour opponent circuitry [1].

4 Introduction para 2

- *Astyanax Mexicanus* specific names should start with a lower case letter
- Why is the scientific name for the bichir not given, when that of the cave fish is?

Response: We appreciate the helpful suggestion. We omitted “*Astyanax Mexicanus*” from this sentence (P3L8).

5 Introduction para 3 this seems to largely repeat table 1. Is this necessary?

Response: As suggested, we removed this paragraph (P3L10).

6 Introduction, para 4 “...fish vision was examined in several ways ...(Table 1)...” Initially I thought this was referring to this study, but it is clear it is referring to previous literature. Although some studies are listed I know it is not an exhaustive list. Salmonid spectral sensitivity, for example, has been widely studied and some rainbow trout studies are not listed. The same is true for goldfish and perhaps other species listed. Is what is listed the study with the longest red sensitivity? If so, this should be clarified.

Response: We appreciate helpful suggestion. This section was rephrased as follows:
(P3L12) Besides OMR, fish vision has been examined in several ways, including electroretinography (ERG), phototaxis, and optokinetic response (OKR). We summarized the longest wavelength to which animals responded (Figure 1, Table S1).

7 The legend of Figure 5 fully explains the figure. However, the legend of the preceding 2 figures (Figures 3&4) gives no such information, limiting their value. In fact, as Figures 3&4 are negative (i.e. show there was no optomotor response). I wonder whether they are needed in the main text at all.

Response: As suggested by Reviewer 1 and Reviewer 2, Figures 3 and 4 were combined into new Figure 3, and we rephrased the legend as follows:

Figures 3 and 4 (New Figure 3) Behavioural response of Nile tilapia and Mexican cavefish under the light from 720 nm to 860 nm.

Behavioural tests were conducted under light of 720, 750, 800, 810, 820, 830, 840, 850, and 860 nm for Nile tilapia (red line), and 720, 750, 800, 820, and 860 nm for Mexican cavefish (black line). In all figures, the horizontal axis indicates the wavelength. The OMR was quantified using the four parameters. (A) Delay. The seconds until a fish turned its body axis and followed the rotating stripes. (B) Duration. The ratio of the time during which a fish followed the stripes divided by the total testing time. A ratio of one means that a fish followed the pattern all the time during the test. (C) Angular Velocity. The speed (degrees/s) at which a fish followed the pattern. (D) Distance. Total Distance (rounds) a fish swam during the test. Values represent means. Error bars are standard errors (SEs).

8 Many of the numerical results given in the text are repetitions of data in tables. This seems unnecessary.

Response: We appreciate the helpful suggestion. We omitted tables 3 and 4. Instead, we mentioned these statistics in text. As suggested by Reviewer 1, we added F/t values in the text.

9 While the data in Figure 5 seems clear, trends in Figure 6 are much less easy to make out.

Response: For clarity, we summarized the graphs according to fish species as in supplementary figures (Figure S1, S2).

10 Figure 7 it is unclear what the blue smudge indicates. The legend say it is the wavelengths which elicited an OMR. However, this cannot be right at shorter wavelengths will have elicited an OMR too. Is it the longest wavelength that elicited an OMR? Why is it a gradient; surely it should be a single value?

Response: As indicated, a smudge was replaced by a bar. In legend of Figure 7 (new Figure 1), we added the following sentence:

(P13L5) The wavelengths of light that have evoked a behavioural or electrophysiological response in fish are depicted as bars and arrows. A bar indicates the longest wavelength that has invoked fish response;

As indicated by Reviewer 1, Figure 7 was combined with Figures 1 and 2 into new Figure 1.

Appendix B

Re: Manuscript ID RSOS-210415

Response to Editor:

Thank you for allowing us to resubmit for the second time. We are also grateful to extend the resubmission deadline. Reviewer 2 made valuable comments on our manuscript, and we revised the paper according to Reviewer2's comment. We have responded to each comment made by Reviewer2.

Response to Associate Editor:

We thank you for allowing us the second resubmission. We are also grateful to extend the resubmission deadline. We have fixed incorrect word and term usage and removed sentences that were not relevant to this study. This paper was checked by professional English editing service. We have addressed all of the concerns of Reviewer2.

The following is a point-by-point response to the questions and comments.

Response to Reviewer 2:

Thank you for your review of our paper. We apologize for not understanding what you meant in the first review. This time we have fixed incorrect word and term usage and removed sentences that are not relevant to this study. The paper was checked by English editing service. We have answered each of your points below. We have incorporated changes that reflect the detailed suggestions you have graciously provided.

Comments to the Author(s)

1 In my original review I said "The authors have chosen to only look at long wavelengths (>720nm). Why they have done this is not explained. Most studies of spectral sensitivity include the whole of the 'visible' spectrum. It must be explained early on why this was not done." In their response letter the authors state that this was due to a "technical issue". Specifically, it seems to concern the well-known phenomenon of fluorescence in the UV. This to me still does not explain why wavelengths between the far red and the UV were not examined. There may well be a good reason for only looking at longer wavelengths, but as I said previously 'It must be explained early on (i.e in the Introduction) why this (examining the whole spectrum) has not been done'. It is an unusual thing to do and needs justification.

The discussion now begins with outlining the problem with UV fluorescence, but as stated above I think this does not justify ignoring wavelengths between 400-700nm. I understand that the value of this study is that it compared different species using the same technique. However, it is still not fully explained why only long wavelengths were studied and most of the visible spectrum was ignored.

Response: We sincerely accepted the comment of Reviewer 2. We added the reason why we examined only long wavelengths in this study. The red colour has an important meaning for fish. This colour triggers aggression in male sticklebacks and is a nuptial colour that affects mating preference in fish. Previously, using *LWS-KO* medaka, we reported that altering red-light sensitivity influenced mate choice. Sensitivity to red light may also be important for other fish species than medaka. Therefore, we examined the red-light sensitivity of several fish species in this study. We omitted the sentences about “technical issue” in DISCUSSION, and added the following sentences in INTRODUCTION:

(P9L1 in first revised manuscript clean version) ~~In OMR assays in the ultraviolet region, we needed to avoid fluorescence contamination (our unpublished observation). Since red light did not have such a problem and stable results could be obtained, we assessed visual sensitivity for long wavelength light.~~

(P2L33 in second revised manuscript clean version) Using a wooden model almost 70 years ago, Tinbergen showed that the colour red triggered aggression in male stickleback fish [18]. Red is also a nuptial colour [19,20] and influences mating preference in fish [21,22]. Previously using medaka with the *LWS* gene knocked out, we showed that changes in red-light sensitivity affect mate choice [23–25]. Red-light sensitivity can also be important for other fish species. To understand the visual sensitivity of several fish species to red light, we conducted an optomotor response (OMR) assay in this study.

2 In my original review I said “This study also claims to look at ‘sensitivity’ at long wavelengths. What the authors did was to determine the response of fish to a set intensity for each wavelength. Much more accurate information could be derived from determining the threshold intensity that evokes a response at each wavelength using a variety of different intensities.” The authors acknowledge this by saying “Determining the threshold intensity at each wavelength, as proposed, may provide us more accurate information about fish vision at each wavelength. However, our result showed that spectral sensitivity for each wavelength were different among fish species.”

While this may be true, it still seems to me that the current data therefore, at best, give an estimate of comparative sensitivity at long wavelengths rather than an accurate measure.

Response: We rewrote the corresponding part according to your suggestion:

(P9L12 in second revised manuscript clean version) (The best speed of stimulus, bandwidth of stripe, and size of apparatus to induce an OMR might vary by fish species. It is possible that the longer the acclimation time, the better the fish response.) We used light with different intensity for each wavelength (Figure 2). Much more accurate information could be derived from determining the threshold intensity that evokes a response at each wavelength using a variety of intensities of light. Furthermore, it is necessary to verify the optimum conditions for inducing the OMR for each fish species such as stripe width, distance from stripes, and rotation speed of striped pattern. In this study, a significant difference between species was shown under consistent conditions. However, the sensitivity of fish OMR measured under various experimental conditions as described above may be different from the results of this study.

3 In my original review I made several comments regarding the first paragraph of the introduction. There seemed to be several inaccuracies. These have now been addressed by the authors, but I feel in doing this the authors have produced a very lengthy paragraph, much of which is not directly relevant to the study. Some specific problems also remain;

3-1 • In my original review I pointed out that the authors had stated there were only 2 types of photoreceptor, to which I commented “This is only true when considering the outer retina. However, the inner retina contains distinct photoreceptive cells (eg melanopsin).” The authors have changed the text so that the paragraph now says there is also a third class of photoreceptor “photosensitive ganglion cells.” This is still inaccurate as other cell types in the retina can also be photosensitive. For example, the first ever non-rod non-cone retinal photoreceptor was described in fish horizontal cells. Also, melanopsin is not the only ‘novel’ photopigment involved.

3-2 • The authors now give quite a lot of information about melanopsin in this first paragraph which does not seem relevant or appropriate to the work presented. I realise that this has probably happened following my original comment, which was only made to point out that the original statement made was wrong. However, I did not mean the authors to include a lot of information that is not really relevant to the current study.

Response to #3: We sincerely accepted the comments that first paragraph of

INTRODUCTION was very lengthy and much of which is not directly relevant to the study. We removed irrelevant statements in the first paragraph and fixed incorrect word and phrase usage.

Response to #3-1 and #3-2: We removed statement about melanopsin in the first paragraph and rewrote as follows::

(P2L19 in second revised manuscript clean version) In the **outer** retinae of vertebrates, there are **two** classes of light-sensitive cells: rod photoreceptors **and** cone photoreceptors, ~~and photosensitive ganglion cells. Photon signals activate one or more types of photoreceptor cells. By comparing the activities of different photoreceptor types, wavelength information has been extracted in colour opponent circuitry [1].~~ Rod photoreceptor cells are responsible for scotopic vision, the vision working under dim light. Photopic vision, the vision working under daylight, is mediated by cone photoreceptors. Colour information is mainly gained by comparing the output of cones [1]. ~~Photosensitive ganglion cells contain the photopigment melanopsin. Melanopsin is involved in the regulation of circadian rhythms and pupil light reflexes [2]. In teleost, such as zebrafish and the African cichlid, melanopsin (OPN4) was identified [3–6]. In humans, melanopsin appears to enhance the perception of brightness [7] and colour [8].,,,,,,~~

3-3 • The first paragraph also refers to “the African cichlid”. This is not very helpful as there are many cichlids in Africa.

Response to #3-3: As suggested, we added “*Astatotilapia burtoni*” (The sentence including “the African cichlid” was moved to P12L13 in DISCUSSION).

3-4 • This first sentence of the first paragraph lists the 3 photoreceptor types, while the third sentence says “By comparing the activities of different photoreceptor types, wavelength information has been extracted...”. This is misleading as it sounds as if to get wavelength information you compare the output of cones, rods and ganglion cells. What should be said is that colour information is (mainly) gained by comparing the output of cones.

Response to #3-4: We removed the sentence “By comparing the activities of different photoreceptor types, wavelength information has been extracted in colour opponent circuitry [1].” And we rewrote as suggested.

(P2L22 in second revised manuscript clean version) Colour information is mainly gained by

comparing the output of cones [1].

3-5 • It is stated that “cones contain one type of visual pigment”. As it stands this statement is very misleading as most retinas contain several distinct spectral cone types and, as the authors later point out, some cones co-express opsins. The statement needs to be changed to ‘individual cones usually contain a single type of visual pigment’. It also has to be made clear that most species have more than one spectral cone type.

Response to #3-5: We corrected this sentence as follows:

(P2L27 in second revised manuscript clean version) Individual cones usually contain a single type of visual pigment.....

Furthermore, we added the statement that “most species have more than one spectral cone type” as follows:

(P2L32 in second revised manuscript clean version) Most fish species have more than one spectral cone type [9–17].

3-6 • Why list the human visual pigments in the introduction to a study on fish?

Thus, in summary, I do not think the introductory paragraph is very helpful. I assume it should give the basic biology needed to understand the study. However, I think much of it is irrelevant and parts of it are misleading.

Response to #3-6: As suggested, we removed the statement about human visual pigments in INTRODUCTION.

4 I understand that in Figure 1B the bars of various colours underneath the x axis indicate the longest wavelength that evoked a visual response using different measures. However, I do not know what the arrows coming off these bars towards the left indicate. Is the length of the arrow indicative of anything?

Response: The arrows mean that animals responded at shorter wavelengths. But the length of the arrow does not indicate anything. We removed arrows in Figure 1B.

5 In the last paragraph of the Introduction previous work on the animals used in this study is discussed. One sentence starts “Guppy responded to light of 600nm” I think it would be helpful to start this sentence with the phrase ‘In previous studies, guppy responded’.

Response: This sentence has been rephrased as suggested (P3L8 in second revised manuscript clean version).

6 We are told in the methods that fish were kept under ‘ordinary’ fluorescent light. This is not enough information. One must also be informed of the intensity (in radiometric units) and the spectrum (or at least details of the lights) as both of these can effect visual pigment expression.

Response: As suggested, we showed the intensity and the spectrum of fluorescent light in Figure 2.

7 In the methods we are told “The time zones during which the experiment was done are as follows: in the first-round assays, 9-11, 750, 900nm; 12-14, 800, 950 nm etc.....”. It is not immediately obvious what this means. I assume what is meant is that the wavelengths 750 and 900nm were tested between 09.00-11.00 etc. If so, could this not simply be stated in this simple manner. Also, this information is only of interest if we know when the light in the maintenance tanks were on and off; we are simply told it was a 14/10hr light dark cycle. This information is important as circadian rhythms could affect spectral sensitivity. Furthermore, although this information is important, I am not sure it is worth describing in the text exactly when what was tested. Perhaps this would be better presented as a table in the supplementary material?

Response: As suggested, the time zones during which the behavioural experiments were performed and the on/off time of the light that illuminated the aquarium are summarized in Supplementary Table S2:

Table S2. The time zones during which the experiment was done.

(S2-1) Cavfish and tilapia

time zone	wavelength(nm)
9-11	720, 750, 860
12-14	800, 810, 820
14-16	830, 840, 850

(S2-2) The first round

time zone	wavelength(nm)
9-11	750, 900
12-14	800, 950
14-16	700, 850, 1000

(S2-3) The second round

time zone	wavelength(nm)
9-11	860
12-14	800, 820, 880
14-16	780, 840

This table summarizes the time zones during which behavioural assays were performed. Cavefish and tilapia were tested as in S2-1. S2-2 showed the timetable in the first round, and S2-3 did so in the second round.

For example, in the first-round assays, the wavelengths 750 and 900 nm were tested between 09.00-11.00. Fish tanks were lit between 7:00-21:00.

8 Virtually no details are given of the optomotor experimental setup. Although unusual, it is a common enough technique for a detailed description to be unnecessary. However, I think some details are required. For example, what angle did the stripes subtend at the eye/retina? Or how wide were the stripes and how far from the eye. Without this information how, for example, can we be sure that the lack of an OMR in some species was not due an inability to discriminate the stripes due to lowered acuity?

Response: We appreciate helpful suggestion. We added following sentences:

(P4L9 in second revised manuscript clean version) (To assess red-light sensitivity, we performed an OMR assay in deep-red and NIR light) based on the procedure previously described [25]. Individual fish were placed in a cylindrical glass tank with a diameter of 18.5 cm. Around the aquarium, a drum with a diameter of 24 cm rotated black and white stripes (2 cm wide). The speed of the rotation was 10 rpm (60°/s). The tank was irradiated from above using monochromatic light generated by an Okazaki Large Spectrograph at the National Institute for Basic Biology (Aichi, Japan) [36]. In the centre of the tank, a 50 ml

centrifuge tube filled with water was placed to prevent fish from short-cutting. As fish swam freely in the aquarium, the distance between fish and the striped pattern changed during behavioural assays, with a minimum of 5.5 cm. (Fish were light adapted for more than.....)

9 Table S1 purports to show the longest wavelength which evoked a response in various species of fish in previous studies. In my previous review I questioned how comprehensive the list of studies was. Obviously, I have not checked all papers for all species studies here, but I think that at least some of the values in table S1 are erroneous. Specifically, in my first review I said “Salmonid spectral sensitivity, for example, has been widely studied and some rainbow trout studies are not listed.” As a specific example, the following study (Douglas, R.H., 1983. Spectral sensitivity of rainbow trout (*Salmo gairdneri*). *Revue canadienne de biologie expérimentale*, 42(2), pp.117-122.) showed a response at longer wavelengths than the 640nm stated in the table for rainbow trout. Similarly, the longest wavelength said to invoke a response in goldfish is 660nm. However, the following paper (Schaerer, S. and Neumeyer, C., 1996. Motion detection in goldfish investigated with the optomotor response is “color blind”. *Vision research*, 36(24), pp.4025-4034.) sees a response at 699nm. I fear I may be misunderstanding what sort of studies are included in the table and why.

Response: In the first revised manuscript, we referred to the paper of Schaerer and Neumeyer in the text (ref. no. #49 in the first revised manuscript clean version) but failed to update the longest wavelength to which goldfish responded. In the second revised version, we corrected this value to 699 nm in the main text, Figure 1 and Table S1. Now we checked via internet and confirmed the information in Table S1. We took screenshots of those papers to which we referred. However, still, there were papers we cannot access. We still cannot read the paper of Douglas R.H, 1983. In light of this situation, we rewrote as follows.: (P3L7 in second revised manuscript clean version) **As far as we know**, we summarized the longest wavelength to which animals responded (Figure 1, Table S1).

10 Regarding Figure 2;

10-1 • It is headed “Spectra of monochromatic light from the Okazaki Large Spectrograph (OLS).” What is meant by ‘from the Okazaki Large Spectrograph’?

Response: ‘from the Okazaki Large Spectrograph’ means “emitted by Okazaki Large Spectrograph, a device that irradiates sequential monochromatic light”. As in #6 of Reviewer 2, we added intensity and spectrum of fluorescent light to Figure 2. Therefore, we changed

this header to “Ambient light and monochromatic light in this work”.

10-2 • The body of the legend say “Spectra were measured separately at $\lambda = 700, 720, 750, 780, 800, 810, 820, 830, 840, 850, 860, 15\ 880, 900, 950,$ and 1000 nm. It is unclear what this means; the spectra are given as solid smooth lines and I assume measurements were taken frequently (e.g. every 1nm). What is written implies they were only taken every 20-30nm.

Response: We conducted OMR assays under the light of 700, 720, 750, 780, 800, 810, 820, 830, 840, 850, 860, 880, 900, 950, and 1000 nm. At all points, the spectra were measured every 1nm and shown differently coloured. We divided these data into three sets: monochromatic light in the OMR assay of cavefish and tilapia, in the first round and the second round. Legends of Figure 2 were rephrased as follows:

(P17L17 in second revised manuscript clean version) (B–D) Spectra of monochromatic light used in the OMR assay. We performed behavioural assays under the light of 700, 720, 750, 780, 800, 810, 820, 830, 840, 850, 860, 880, 900, 950, and 1000 nm. At all points, the spectra were measured every 1 nm and are shown differently coloured. Spectra of light used in OMR assays of cavefish and tilapia (B), the first round (C) and the second round (D) are illustrated.

10-3 • The legend goes on to say “The different photon density from the previous paper [23] depended on the cumulative lighting time of the xenon lamp, the light source.” I am afraid I do not know what this means.

Response: This was a response to the comment #4 of Reviewer 1 (“I notice many of the wavelengths are similar to those described in previous work (i.e., Matsou et al. 2018) but the photon densities differ.”). We moved this sentence to METHOD and rephrased as follows:

(P4L21 in second revised manuscript clean version) The photon density used in this study (Table 1) was different from our previous paper [25] because the photon density changed depending on the cumulative lighting time of the light source (the xenon lamp).

11 The legend of Figure 3 describes the 4 parameters measured in detail. A very similar description is given at the end of section 2.2. Is such repetition necessary? I realise this is most likely a response to my previous comment that the legend was not descriptive enough.

Response: We omitted the definition of four parameters. Instead, we inserted “(refer to “METHOD, section 3.2” for details of the four parameters)”

12 Section 3.1, 2nd para states that rainbow trout and tilapia turned around frequently and refers us to Fig. 3. As far as I can tell, Fig. 3 does not allow us to detect a fish turning.

Response: We changed “Figure 3” into “Video file S1” as follows:

(P6L5 in second revised manuscript clean version) For example, young rainbow trout and Nile tilapia turned around frequently during the OMR test, regardless of the rotating stripes (Video file S1).

We added Video file S1 recording behavioural test of rainbow trout.

13 The following section (3.2) explains well how the data show cavefish do not have an OMR. It might be sensible to merge sections 3.1 & 3.2 and describe all the species that do not have an OMR together?

Response: We rewrote these sections as suggested:

(P6L2 in second revised manuscript clean version) **4.1. Five fish species did not have an OMR**

First, we performed an OMR assay at 700 nm or 720 nm. The response of four fish species (rainbow trout, Nile tilapia, Senegal bichir and Japanese striped loach) suggest they should be evaluated for their photosensitivity by other methods. For example, young rainbow trout and Nile tilapia turned around frequently during the OMR test, regardless of the rotating stripes (Video file S1). Japanese striped loach and Senegal bichir seldom swam and did not follow the stimuli. Yet, the bichir moved its head whenever we switched the direction of stripe rotation (Video file S2), suggesting that it perceived light even though the OMR was negative.

Mexican cavefish have degenerated eyes. We tested their behaviour under light of wavelengths 720, 750, 800, 820, and 860 nm. Values of the four parameters were computed and are summarized in Figure 3. Under all wavelengths, the Delay parameter fluctuated and had no significant difference (one-way repeated measures ANOVA, $F(4, 20) = 0.6431$, $P = 0.6381$). Duration was between 0.202 (750 nm) and 0.304 (820 nm) ($P > 0.05$ in all comparisons). From 720 nm to 860 nm, Angular Velocity was maintained (one-way repeated measures ANOVA, $F(4, 20) = 0.7193$, $P = 0.5888$). Distance at any wavelength was negative, which meant cavefish never followed the signals (one-way repeated measures ANOVA, $F(4,$

20) = 0.5802, P = 0.6804). Based on these four parameters, we judged cavefish to be OMR negative at all wavelengths tested.

4.2. Eight fish species were OMR positive

Based on our manual observations, eight fish species (medaka, goldfish, zebrafish, guppy, three-spined stickleback, mbuna, glowlight tetra, and bronze corydoras) were OMR positive (examples of the OMR assay of tetra and corydoras are in Video files S3 and S4). Among them, six fish species (medaka, goldfish, zebrafish, guppy, three-spined stickleback, and mbuna), widely used as model animals, were subjected to further behavioural assays.

14 Figure 4 legend; The title of the legend is “Behavioural response of medaka, goldfish, zebrafish, guppy, three-spined stickleback, and mbuna under the light at 50 nm intervals from 700 nm to 1000 nm”. The first line of the legend itself begins “Behavioural tests were conducted under light of 700, 750, 800, 850, 900, 950, and 1000 nm using 34 goldfish, zebrafish, three-spined stickleback, and mbuna.” These 2 sentences seem to say the same thing. The same comment applies to the legend of figure 5.

Response: As suggested, we changed the title of Figure 4 and 5 as follows:

Figure 4. Behavioural response of medaka, goldfish, zebrafish, guppy, three-spined stickleback, and mbuna in the first-round OMR assay.

Figure 5. Behavioural response of medaka, goldfish, zebrafish, guppy, three-spined stickleback, and mbuna in the second-round OMR assay.

15 Previously I had said “Many of the numerical results given in the text are repetitions of data in tables. This seems unnecessary.” In response to this the authors have removed the tables and reported all the numerical data in the text. I find this makes the text very hard to read and follow. The second paragraph of section 3.3, for example, summarises the overall findings depicted in figure 4 well. The following three paragraph, however, add little to this understanding. I therefore wonder if rather than remove the table, as the authors have done, what should have been done is to keep the tables but remove the details from the text.

16 Similarly, I think section 3.4 would benefit from less numerical data in the text and should concentrate on a summary of the long wavelength limits of the OMR in the various species.

Response to #15 and #16: We appreciate the helpful suggestion of Reviewer 2. However,

Reviewer 1 suggested adding statistics in the main text. For instance,

#7 P6L31-41: Please provide statistics/p-values supporting the statements herein.

#8 P7L25: p-values are used but please report the F-statistic or any other appropriate statistic for the parametric/non-parametric tests used.

#44 P8L26: Report the p-values and F-statistics.

Both comments of Reviewers 1 and 2 were valuable, so we rewrote the text with less numerical data and kept some statistical values described. We also added the statistics as Tables 2 and 3.

16 Throughout the discussion there is confusing and incorrect use of the terms chromophore and opsin. A visual pigment is composed of a light-absorbing chromophore linked to a protein, opsin, which tunes the chromophore. Neither opsin nor the chromophore are a visual pigment. Thus, the Discussion says “The transition from A1 to A2 results in a red shift of spectra which opsins absorbs”. The opsin does not absorb light. It is a protein that tunes the light absorbing chromophore. The sentence should probably read something like ‘The transition from A1 to A2 results in a red shift of the visual pigment’s absorption spectrum’. Later it says “... these opsins could absorb light of longer wavelengths. Once more, the opsin does not absorb the light, the visual pigment does. Also “retinal A2 works as the visual pigment...” is incorrect as retinal is not the visual pigment. There are several other examples of similar confusions

Response: We rewrote corresponding part as follows:

(P10L3 in second revised manuscript clean version) The transition from A1 to A2 results in a red shift of the visual pigment’s absorption spectrum [14,45].

(P11L32 in second revised manuscript clean version) LWS pigment of these fish could absorb light of longer wavelength.

(P11L34 in second revised manuscript clean version) retinal A2 works as a chromophore in vivo [14,45].

In addition, we fixed incorrect word and phrase in the text.

17 Section 4.2 begins with “In past reports, zebrafish and guppy have shown phototaxis under light of wavelength >900 nm, but they did not respond to light >800 nm in the OMR assay.” References need to be provided to substantiate these claims. A few sentences later the authors say “The reason why goldfish could perceive light of longer wavelength is not

clear". It needs to emphasise that this sentence refers to 'in this study'. Without these insertions it is insufficiently clear when previous studies are being referred to and when statements refer to the present work.

Response: As suggested, we rephrased corresponding sentences as follows:

(P10L19 in second revised manuscript clean version) In past reports, zebrafish and guppy have shown phototaxis under light of wavelength >900 nm [34], but they did not respond to light > 800 nm in the OMR assay [26,28].

(P10L22 in second revised manuscript clean version) The reason why goldfish could perceive light of longer wavelength **in this study** is not clear.

18 At the end of the Discussion there is mention of the role of melanopsin in the OMR. I just want to repeat what I said at the beginning of this review, melanopsin is not the only non-rod, non-cone visual pigment in fish. There is, for example, VA opsin in horizontal cells.

Response: We rephrased the corresponding part in the last paragraph of DISCUSSION as follows:

(P12L10 in second revised manuscript clean version) (,,,, after light adaptation.) Non-cone, non-rod visual pigments may be involved in the perception of red light. For example, in humans, melanopsin is involved in the perception of brightness and colour [56,57]. In teleost, such as zebrafish and the African cichlid (*Astatotilapia burtoni*), melanopsin was identified [58–61]. Other non-cone, non-rod visual pigments are also candidates in red-light perception [62–67]. Future studies testing the OMR using fish with knock-out LWS, rhodopsin genes or non-cone, non-rod visual pigments component genes will solve this question.

Response to Reviewer 1

Thank you for providing us with many suggestions and advice. We are also grateful for the time and energy Reviewer 1 expended. By virtue of the valuable comments of Reviewer 1, our paper has become much better.

Reviewer: 1

Comments to the Author(s)

The revised manuscript by Matsuo et al. entitled "Behavioural red-light sensitivity in fish according to the optomotor response" is a valuable addition to the field and in its current state is suitable for publication. The authors conscientiously addressed my critical and rather lengthy comments from the first round of review and I appreciate their efforts. I particularly appreciate the lengths the authors went to clarifying the methods and making them more reproducible, the added sections on ontogeny and chromophores added interesting context, and the figures are much more informative. Figure one in particular is a marked improvement and conveys a lot of really useful details -- my one comment would be to include the data reported here in Fig 1, not just previously reported behavioural and ERG data from the literature (e.g., goldfish are shown to have OMR from wavelengths approximately <650nm but the authors report much longer wavelengths eliciting an OMR response). Overall, I think these data are reported well in the manuscript and publication in the Royal Society Open Science is justified.

Appendix C

Re: Manuscript ID RSOS-210415

Response to Editor:

Thank you for allowing us the resubmission. Reviewer 2 made valuable comments on our manuscript, and we revised the paper according to Reviewer 2's suggestions. We have responded to each comment made by Reviewer 2.

Response to Associate Editor:

We thank you for giving us the opportunity of resubmission. We appreciate many valuable suggestions and corrections made by Reviewer 2. We have addressed all of the concerns of Reviewer 2.

The following is a point-by-point response to the questions and comments of Reviewer 2.

Response to Reviewer 2:

1 Figure 1 legend this refers to upper and lower columns. Columns are vertical structures. What is meant is 'rows'. The last sentence of the legend says "Previous works on opsin genes and fish vision are summarised in Supplemental Table S1". When I first read this I was confused as to what this table might contain as 'fish vision' is rather a big subject. I did not immediately understand that only the contents of this figure was being referred to. I therefore suggest this sentence be replaced by something like "The original references on which this figure is based are given in Supplemental Table S1"

Response: We are sorry for confusing columns and rows. We corrected these words and the last sentence of Figure 1 legend accordingly.

2 Figure 2 Panels (B)-(C) are headed "Wavelengths conducted". I do not think 'conducted' is the correct word. Perhaps 'used' would be more appropriate.

Response: We fixed 'conducted' to 'used' in Figures 2-B, C and D.

3 In the Methods the length of the fish is given as 3-10cm. When specifying fish size one usually uses 'standard length' (snout to caudal peduncle) but 'total length' (which

includes the caudal fin) is also sometimes used. It should be specified whether SL or TL is being used here.

Response: We measured fish size from a snout to a caudal fin. We clarified this point in P4L3 as follows:

(P4L3) We measured the total length (TL) of the fish from the snout to the caudal fin. All the fish were 3–5 cm long

4 Thank you for including information about the time-of-day various experiments were done in Table S2, as this is potentially important information. However, I have one more minor suggestion. In the text you refer to “time zones”. In English this has a very specific meaning and refers to the time of day at various points on the planet. Of course, this is not what you mean. It would be much clearer for the reader if you were to simply refer to the time-of-day experiments were done and avoid the phrase ‘time zone’.

Response: We now understand the “time zones” means the time of day at various points on the planet. The phrase “time zones” was removed, and we rewrote the corresponding part as follows:

(P5L1) The approximate schedule of experiments is described in Table S2.

(P22L10) The schedule of experiments.

(P22L21) This table summarizes the time-of-day experiments were done.

5 I appreciate you have taken a detailed description of the parameters out of the legend of Figure 3. In truth I don’t think it is necessary to refer to the Methods for details of the stimuli as this is self-evident and not normal practice. I also do not think you need to say which wavelengths were tested in the legend either as it is obvious from the figure and is also outlined in the Methods. Similar comments apply to the legends of Figs. 4 & 5.

Response: In the legends of Figures 3, 4 and 5, we rewrote as follows:

(P18L24, in the legend of Figure 3) Behavioural tests were conducted ~~under light of 720, 750, 800, 810, 820, 830, 840, 850, and 860 nm for using Nile tilapia (red line), and 720, 750, 800, 820, and 860 nm for Mexican cavefish (black line).~~

(P18L30, in the legend of Figure 4) Behavioural tests were conducted ~~under light of 700, 750, 800, 850, 900, 950, and 1000 nm using six fish species. –goldfish, zebrafish, three-~~

~~spined stickleback, and mbuna. Medaka and guppy were tested under light of 700, 750, 800, 850, and 950 nm.~~

~~(P18L36, in the legend of Figure 5) Behavioural tests were conducted under light of 780, 800, 820, 840, 860, and 880 nm using goldfish, zebrafish, three-spined stickleback, and mbuna six fish species. Guppy were tested under light of 780, 820, 840, 860, and 880 nm. OMR assays using medaka were performed under light of 780, 800, 820, 840, and 880 nm.~~

6 Tilapia data are shown in Figure 3, and this is referred to in section 5.2 of the Discussion. Would it be worthwhile to also refer to the tilapia data in Fig. 3 in the first section of the results (4.1)?

Response: Yes, it would be worthwhile. According to Reviewer 2's suggestion, we added the following sentences in 4.1 of the Results. To avoid repetition, we rephrased the last two sentences in 5.2 of the Discussion as follows:

~~(P6L20, in 4.1) Nile tilapia did not exhibit an obvious OMR. However, Delay increased towards 860 nm, Angular Velocity was about 60°/s, and Duration and Distance fluctuated but never declined (Figure 3). When comparing the behavioural response of cavefish and tilapia, Nile tilapia could perceive light of wavelengths between 720 nm and 860 nm.~~

~~(P10L32, in 5.2) In this work, Nile tilapia did not exhibit an obvious OMR (Figure 3). Delay increased towards 860 nm, Angular Velocity was about 60°/s, and Duration and Distance fluctuated but never declined. Based on four OMR these parameters, tilapia could perceive be OMR positive under the light of wavelengths of 720 nm to 860 nm.~~

7 I still find sections such as 4.3 difficult to read due to the plethora of numerical data and statistics in the text. The data in Fig 4 are clear and one can see that in all species 'duration' increases with increasing wavelength while the other 3 parameters fall, indicating OMR is lost at longer wavelengths. For me, a simple statement along these lines would suffice, rather than the complex text presented in this section. This is especially true for this section, which is only exploratory for the subsequent experiments. Essentially similar comments apply to the following section (4.4). However, I accept the authors have a problem if presenting the statistics in this way was requested by the other reviewer and will accept what they have done.

Response: Because we are hesitant to ignore comments from Reviewer 1 (who has already accepted these changes), we somehow reduced the numerical data by not describing those when a significant difference was not detected. We hope this meets Reviewer 2's and Editor's approval.

8 In the Discussion it says "As the light source used in this study produced a sequential wavelength of light, a bigger apparatus and a larger aquarium covered a wider wavelength range." I do not understand what this means. Why does a bigger aquarium result in a larger wavelength range?

Response: The spectrograph produced a sequential series of monochromatic light of wavelength 250 nm to 1000 nm and irradiated it on a U-shaped focal surface of about 10 meters wide. This means that an apparatus of 10 m in diameter receives the light of all wavelengths (i.e., from 250 nm to 1,000 nm) and that an apparatus of 5 m in diameter receives about half of all the wavelengths (e.g., from 250 nm to 625 nm, or from 500 nm to 875 nm). That is, the smaller the apparatus was, the purer monochromatic light it could receive.

We rewrote the corresponding part as follows:

(P10L2) Our spectrograph horizontally dispersed a series of monochromatic lights of wavelength 250 nm to 1000 nm on a U-shaped focal surface of about 10 meters wide. Hence, a bigger apparatus and a larger aquarium covered a wider wavelength range.

9 The Discussion says "In goldfish, LWS-expressing cones process stimuli to elicit an OMR under light of wavelengths between 409 nm to 699 nm while in adult zebrafish, stimulus occurs under light of 416 nm through to 699 nm [27,28]." It is unclear what this means. The term 'stimulus occurs' is especially confusing. Is the following what is meant 'In goldfish, LWS-expressing cones elicit an OMR at wavelengths between 409-600nm while in adult zebrafish an OMR occurs between 416-699nm'.

Response: Yes. We corrected this sentence according to Reviewer 2's suggestion (P11L15).

10 The Discussion says 'The longest wavelength that provoked an OMR in medaka,

goldfish, zebrafish, guppy and stickleback was 800 nm, while being under 780 nm in mbuna. In medaka, the absorption spectrum of LWS pigment was almost zero at 650 nm [9]; in stickleback, 650 nm [10]; in Nile tilapia, 675 nm [11]; and in guppy, 650 nm [12]. The authors seem concerned that their fish still respond at wavelengths where the absorption of the visual pigment appears to zero. This is, however, not really a problem as the absorption spectrum only seems to approach zero at the wavelengths stated as the curves the visual pigment spectra shown in Fig. 1 are drawn on a linear scale. However, photoreceptors respond over a range on many log units and if visual pigments are displayed using log absorbance it is apparent they could easily mediate longer wavelength sensitivity.

Response: We appreciated the suggestion of Reviewer 2. We rewrote accordingly. (P12L23) Alternatively, the gap may reflect photoreceptors' ability to respond over a range on many log units. If visual pigments are displayed using log absorbance, they could easily mediate longer wavelength sensitivity.

11 In the last paragraph of the Discussion the authors mention melanopsin may mediate long wavelength sensitivity. This is very unlikely as across all vertebrates melanopsin is maximally sensitive at around 480nm.

Response: In response to Reviewer 2's comment, we stated that melanopsin is unlikely to mediate long-wavelength sensitivity.

(P12L20) However, as melanopsin is maximally sensitive at around 480nm [59,62], it is very unlikely that melanopsin mediates long-wavelength sensitivity.